# Combining genetic and demographic monitoring better informs conservation of an endangered urban snake

**Dustin A. Wood** [1]*, **Jonathan P. Rose**[2], **Brian J. Halstead**[3], **Ricka E. Stoelting**[4], **Karen E. Swaim**[4], **Amy G. Vandergast**[1]

**1** U.S. Geological Survey, Western Ecological Research Center, San Diego Field Station, San Diego, California, United States of America, **2** U.S. Geological Survey, Western Ecological Research Center, Santa Cruz Field Station, Santa Cruz, California, United States of America, **3** U.S. Geological Survey, Western Ecological Research Center, Dixon Field Station, Dixon, California, United States of America, **4** Swaim Biological Incorporated, Livermore, California, United States of America

* dawood@usgs.gov

## Abstract

Conversion and fragmentation of wildlife habitat often leads to smaller and isolated populations and can reduce a species' ability to disperse across the landscape. As a consequence, genetic drift can quickly lower genetic variation and increase vulnerability to extirpation. For species of conservation concern, quantification of population size and connectivity can clarify the influence of genetic drift in local populations and provides important information for conservation management and recovery strategies. Here, we used genome-wide single nucleotide polymorphism (SNP) data and capture-mark-recapture methods to evaluate the genetic diversity and demography within seven focal sites of the endangered San Francisco gartersnake (*Thamnophis sirtalis tetrataenia*), a species affected by alteration and isolation of wetland habitats throughout its distribution. The primary goals were to determine the population structure and degree of genetic isolation among *T. s. tetrataenia* populations and estimate effective size and population abundance within sites to better understand the present and future importance of genetic drift. We also used temporally sampled datasets to examine the magnitude of genetic change over time. We found moderate population genetic structure throughout the San Francisco Peninsula that partitions sites into northern and southern regional clusters. Point estimates of both effective size and population abundance were generally small ($\leq 100$) for a majority of the sites, and estimates were particularly low in the northern populations. Genetic analyses of temporal datasets indicated an increase in genetic differentiation, especially for the most geographically isolated sites, and decreased genetic diversity over time in at least one site (Pacifica). Our results suggest that drift-mediated processes as a function of small population size and reduced connectivity from neighboring populations may decrease diversity and increase differentiation. Improving genetic diversity and connectivity among *T. s. tetrataenia* populations could promote persistence of this endangered snake.

**Data Availability Statement:** The ddRADseq data used for this study are available from the NCBI Sequence Read Archive: BioProject PRJNA608966 (https://www.ncbi.nlm.nih.gov/sra/PRJNA608966).

All data that pertain to demographic analyses are included in an R file (includes the code and data) available at U. S. Geological Survey Science Data Catalog (https://doi.org/10.5066/P9YKLBB5)." Specific site locality information are private for this endangered subspecies but can be accessed from the U. S. Geological Survey Science Data Catalog (https://doi.org/10.5066/P9YKLBB5) once individuals or entities have submitted a request and have been approved.

**Funding:** Funding for this project was provided by U.S. Geological Survey, Western Ecological Research Center and Ecosystems Mission Area and U.S. Fish and Wildlife Service, Sacramento Fish and Wildlife Office. The funders had no role in study design, data collection and analysis, decision to publish, or preparation of the manuscript.

**Competing interests:** Swaim Biological, Inc collected Thamnophis sirtalis tetrataenia tissues between the years 2005 – 2015 that were provided as an in-kind service and these tissues were utilized in this study. Site data from a previously published study [35], which was partially funded by Swaim Biological, Inc, were also used in the present study. This does not alter our adherence to PLOS ONE policies on sharing data and materials.

## Introduction

Contemporary expansion of urban environments (starting in the mid-20$^{th}$ century) and associated infrastructure into natural areas has been a major cause of habitat loss and fragmentation of wildlife habitat [1, 2]. As a consequence, wildlife populations often become restricted to smaller, isolated patches that are more vulnerable to extirpation [3, 4]. When connectivity across a species' range is disrupted, the levels and distribution of genetic diversity across local populations can deteriorate as a result of genetic drift (chance loss of alleles through time), which can lead to reduced adaptive potential [5–7]. Genetic diversity provides the raw material for natural selection and is governed in part by several demographic determinants such as population size, gene flow and life history characteristics [8]. In fragmented landscapes, these demographic determinants can influence the magnitude of genetic drift within local populations. Loss of genetic diversity can be rapid when population sizes are small and lack gene flow from adjacent populations, increasing the frequency of inbreeding and resultant exposure of recessive deleterious mutations that can escalate a population's extinction risk [9–11]. In contrast, larger population sizes and increased immigration and gene flow can counter the loss of genetic variation and promote population persistence [8, 12]. For species of conservation concern, monitoring and quantifying parameters related to local population size and connectivity across a species' range provide crucial information to better manage isolated populations and implement effective mitigation measures to maintain species viability [13].

In conservation biology, genetic effective size ($N_e$) of a local population, rather than census size, can be used to measure the influence of genetic drift and provides a way to quantify the amount of genetic diversity that can be maintained in a population. This parameter is defined as the number of individuals in an idealized population that would show the same amount of genetic diversity as the population being measured [14]. Effective population size often deviates from the census size, as an idealized population excludes demographic determinants that increase genetic drift in real populations, such as nonrandom reproductive success, unequal sex ratios, inbreeding and changes in population size. These demographic changes may result in a reduction in $N_e$, and therefore, an increased loss of genetic diversity [11]. In this way, estimates of $N_e$ measure the ability of vulnerable populations to maintain genetic diversity in future generations [14] and, when $N_e$ is small, can prompt investigation into demographic factors that might result in low $N_e$ estimates [15, 16]. When possible, estimating both effective size ($N_e$) and census size ($N$) can provide a robust understanding of population viability [17]. One common approach is to evaluate the ratio $N_e/N$ across populations, given the direct connection of demographic and genetic processes represented in the ratio [18, 19], to help determine possible changes to local populations and inform management decisions.

The San Francisco Peninsula of California, USA has experienced substantial loss and modification of natural habitats over the past century as a result of urban and agricultural development [20, 21]. This region is home to the San Francisco gartersnake, *Thamnophis sirtalis tetrataenia*, which is endangered under both the U.S. and California Endangered Species Acts [22, 23] and classified as imperiled by NatureServe [24]. Although historically common, this species persists in the remnant and highly fragmented coastal marshes, wetlands, and forests of the San Francisco Peninsula from San Mateo County to northwestern Santa Cruz County, California, USA. Primary threats and impacts to *T. s. tetrataenia* survival have been alteration and isolation of habitats resulting from continued expansion of urbanization, decline of prey, and harvesting pressure by collectors. Six populations of *T. s. tetrataenia* are currently the focus of conservation efforts [25], although additional populations might occur on private lands. Recovery criteria include protecting and maintaining 10 populations with a minimum of 200 adults in a 1:1 sex ratio [26]. At present, too little is known about the size of local

populations and the distribution of genetic diversity among them to adequately assess strategies for maintaining maximum genetic diversity within and across populations. Re-establishing or augmenting *T. s. tetrataenia* populations may increase the likelihood of their long-term persistence as additional populations would safeguard against extinction if one or more of the existing populations were extirpated due to stochastic events (e.g., wildfire, saline inundation of marsh habitat, and disease). In addition, captive breeding and translocations can play a vital role in the recovery of a species by providing a "rescue" mechanism to populations under immediate threat [27, 28]. Before establishing captive breeding and translocation programs, consideration of key factors is prudent: (i) how many founding populations would be necessary to capture the population structure and diversity present in the wild, (ii) which wild "source" populations have large and stable abundance and could best tolerate loss of individuals (for captive breeding or translocation efforts) without compromising viability [29, 30], and (iii) which life stages should be taken from source populations to minimize effects to those populations and conversely, which life stages could be used to most effectively establish or augment populations [31].

In this study, we used genome-wide SNP data and capture-mark-recapture models to provide estimates of effective size ($N_e$) and abundance ($N_a$; an approximate estimation of census size ($N$)) to evaluate the population genetic diversity and size across the northern, central and southern range of *T. s. tetrataenia* along the San Francisco Peninsula. We use these two datasets to make conservation-relevant inferences about genetic and demographic characteristics of populations across the range of *T. s. tetrataenia*. The objectives of this study were to (1) determine the degree of population structure and genetic isolation among populations of *T. s. tetrataenia* using the genome-wide SNP dataset, (2) estimate current levels of $N_e$ and $N_a$ within seven focal sites to better understand the present and future importance of drift in this subspecies; (3) use temporally sampled datasets to determine if there is evidence of genetic change over time, and (4) identify source populations that could be used in genetic rescue and captive breeding efforts, and conversely, populations that could benefit from genetic rescue. This study is a comprehensive genetic and demographic analysis of populations of *T. s. tetrataenia* and provides information that could be used as a basis for population management that is consistent with natural patterns of diversity across the range of *T. s. tetrataenia*.

## Materials and methods

### Field methods and sample collection

Between 2016–2018, we performed demographic surveys at seven sites (hereafter focal sites) that span the geographic range of *T. s. tetrataenia* (Fig 1). Four focal sites were located in northern San Mateo County: Pacifica, Skyline, Crystal Springs, and San Bruno. The remaining three focal sites were located in central and southern San Mateo County: Mindego, Pescadero, and Año Nuevo. During demographic surveys we also collected tissue samples for genetic analysis by removing a 5–10 mm tail tip from each snake and immediately preserved the tissue in 95-percent ethanol. At Pacifica, San Bruno, Pescadero, and Año Nuevo, tissue samples were also available from surveys conducted between 2004–2010 that were used in combination with the 2016–2018 tissues to investigate genetic change over time. We also obtained tissues from five sites (hereafter satellite sites; Fig 1) with small sample sizes (2 to 7 per site) that we included in phylogenetic and population structure analyses only.

We sampled Pacifica, Skyline, Crystal Springs, Pescadero, and Año Nuevo from 5 April to 8 June 2018, for 29–46 days per site per year (Table 1). At Mindego we sampled for 53 days from 2 April to 24 May 2016, and at San Bruno we sampled for 68 days from 3 April to 9 June 2017. We constructed 8–12 m long and 0.4 m high drift fences from tempered hardwood boards and

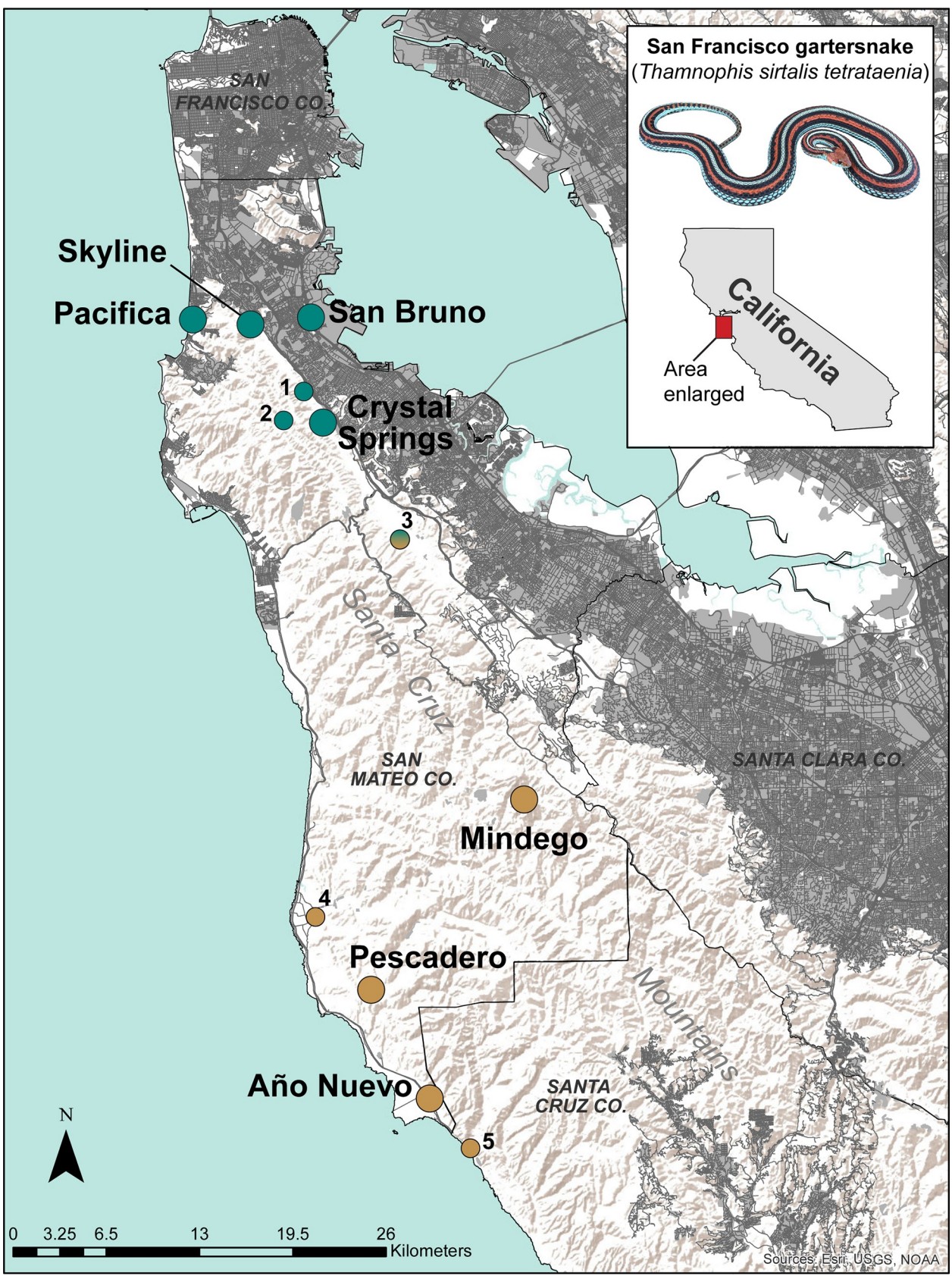

**Fig 1. Sampling map and cluster assignments at K = 2 across the San Francisco Peninsula in California, USA, with the focal sites (larger dots) and satellite sites (smaller, numbered dots) that were used in rangewide and phylogenetic datasets.** Light grey areas and dark grey lines respectively correspond to urban areas and major highways/streets (World Terrain Basemap source: USGS, ESRI, NOAA. Urban land coverage for San Francisco Bay Region source: Bay Area Open Space Council, GreenInfo Network, Conservation Lands Network, and San Francisco Bay Area Upland Habitat Goals Project. (2011). California Urban Lands: Farmland Mapping and Monitoring Project, 2006. Bay Area Open Space Council. Available at http://purl.stanford.edu/kh450fm7856).

installed them, buried approximately 5–7 cm into the soil, adjacent to wetlands and in nearby upland habitat. Each fence had four rectangular wooden funnel traps measuring 30 cm x 40 cm x 23 cm with a hardware cloth funnel facing towards the center of the fence. Two funnel traps were placed flush to each side of the fence at either end [32]. We installed 6–84 drift fences per site, resulting in 24–336 funnel traps deployed per site (Table 1). We checked traps once per day during the afternoon (starting at approximately 1400–1500 hours) at Año Nuevo, Pescadero, and Pacifica. At Crystal Springs, Mindego, San Bruno and Skyline we checked traps twice each day, once in the morning (starting between 0830 and 1000) and once in the afternoon (starting at or after 1400). At Pescadero, we also deployed 101 artificial cover objects (52 2 ft x 3 ft wooden boards, 49 2 ft x 3 ft corrugated metal) to aid in capturing snakes, following long-term survey protocols at this site. At all sites we made hand captures of *T. s. tetrataenia* when possible.

Sites differed in size of available habitat and in the area sampled by traps and cover objects. To define the total area of available habitat for each site, we created polygons in ArcGIS version 10.7.1 [33] that encompassed all suitable habitat, whether wetlands or non-forested uplands. The effective area sampled was then calculated by using a fixed buffer of 200 m around all trap and artificial cover object locations for each site. We chose a 200 m buffer based on the maximum distance moved between captures for greater than 95 percent of individual *T. s. tetrataenia* at our study sites (S1 Fig). The total site area and effective area sampled for each site are given in Table 1. At Pacifica and San Bruno, nearly all of the available habitat for *T. s. tetrataenia* was within the area effectively sampled by traps, and the nearest known population of *T. s. tetrataenia* was more than 2.5 km away. In contrast, although the area sampled at Skyline and Crystal Springs was comparable to Pacifica, only 60–70 percent of the area was sampled because suitable wetland habitat was present nearby and additional habitat (not included in our calculations) is present along most of the 7 km corridor separating these two sites. Año Nuevo, Mindego, and Pescadero were much larger sites and the area sampled was less than 60 percent of the available habitat.

We identified all captured snakes to species and measured snout-vent length (SVL) and tail length to the nearest millimeter using a meter stick. We measured the mass of each captured snake in grams using a Pesola® spring scale and determined sex by cloacal probing. We were

**Table 1. Summary of field efforts at the seven focal sites including number of traps, duration of trapping effort, number of trap nights, captures, number of individuals (# Ind), number of females (F) and males (M), and unknown sex (U), total area (TA) of available habitat and effective area sampled (EAS) for each site.**

| Site | # Traps | Start Date | End Date | Trap-nights | Captures | # Ind | F | M | U | TA (km²) | EAS (km²) |
|---|---|---|---|---|---|---|---|---|---|---|---|
| Pacifica | 48 | 4/19/2018 | 5/24/2018 | 1680 | 47 | 25 | 16 | 9 | 0 | 0.28 | 0.20 |
| Skyline | 24 | 5/8/2018 | 6/8/2018 | 696 | 27 | 24 | 16 | 6 | 2 | 0.34 | 0.20 |
| Crystal Springs | 24 | 5/8/2018 | 6/8/2018 | 696 | 27 | 22 | 14 | 7 | 1 | 0.30 | 0.21 |
| San Bruno | 336 | 4/3/2017 | 6/9/2017 | 22336 | 727 | 555 | 239 | 262 | 0 | 0.73 | 0.70 |
| Mindego | 48 | 4/2/2016 | 5/24/2016 | 2544 | 121 | 86 | 30 | 55 | 1 | 1.93 | 0.98 |
| Pescadero | 96 | 4/7/2018 | 5/23/2018 | 4416 | 60 | 51 | 23 | 26 | 2 | 3.07 | 1.82 |
| Año Nuevo | 48 | 4/5/2018 | 5/21/2018 | 2208 | 79 | 52 | 24 | 27 | 1 | 1.65 | 0.53 |
| Total | 624 | | | 34576 | 1088 | 815 | 362 | 392 | 7 | | |

unable to determine the sex of a few snakes because wounds near the vent or tail prevented probing. At all sites except San Bruno, we gave each *T. s. tetrataenia* a unique ventral brand with a medical cautery unit that corresponds to a numerical code [34]; at San Bruno, we marked snakes with unique ventral scale clips and also implanted each with Passive Integrated Transponder (PIT) tags. We processed and released all snakes at their site of capture within one hour. Snakes were handled in accordance with IACUC Protocol WERC-2015-01, which was approved by the Western Ecological Research Center Animal Care and Use Committee in association with the University of California, Davis, and as stipulated in U.S. Fish and Wildlife Service Recovery Permits TE157216-4 and TE815537-10, California Department of Fish and Wildlife Scientific Collecting Permits/MOUs SC- 10779 and SC-002672.

## Capture mark recapture modeling and census size estimation

We used population abundance estimates ($N_a$) of adult *T. s. tetrataenia* to approximate census size ($N$) for each site during its respective survey period. We used a size threshold of 300 mm SVL to classify individuals as mature adults, following [35]. At all sites except San Bruno, we used a Bayesian multinomial N-mixture model [36] with random effects of date, site, and individual on capture probability ($p$) to estimate $N_a$ for the five sites monitored in 2018. By treating site as a random effect, this multi-site model shares information about the capture process among sites, allowing for more precise estimates of capture probability and abundance at each site. We also included fixed effects of air temperature, sex, SVL, and a behavioral response to previous capture on capture probability. The behavioral response tests whether there is an effect of being captured on day *d-1* on the probability of being captured on day *d*. We centered and standardized the air temperature and SVL covariates by subtracting the mean from each measured value and dividing by the standard deviation to produce covariates with a mean = 0 and a standard deviation = 1. We evaluated the importance of covariates on capture probability by calculating the number of model iterations in which the parameter had the same sign as the median estimate and dividing by the total number of iterations. For example, if the median estimate for the effect of air temperature was positive, we calculated the posterior probability of a positive relationship as the proportion of posterior samples in which the air temperature coefficient was positive. We considered evidence for a positive (or negative) relationship between a covariate and capture probability to be strong if the posterior probability of the parameter being positive (or negative) was $\geq$ 0.9. For Mindego, we used the same model, except that no random effect of site was included because the model was fitted to data for a single site. For San Bruno, we used the similar Huggins closed capture modeling approach [37, 38] with results reported from a model with sex and random time effects retained for capture probability.

We estimated the number of *T. s. tetrataenia* that were present but not captured each year using data augmentation [39]. At all sites except Mindego and San Bruno, we augmented the observed capture histories of San Francisco gartersnakes with enough additional, all-zero capture histories for a total pool of 1000 individuals over all five sites. The model then estimated how many of these 1000 individuals were present and available to be captured, at each site, and this was the estimated abundance of *T. s. tetrataenia*. For Mindego, we augmented the observed capture histories to create a total pool of 500 individuals, and at San Bruno, we augmented the observed capture histories with an additional 2000 individuals, for a total pool of 2555 individuals. We fit Bayesian capture-mark-recapture (CMR) models using the software Just Another Gibbs Sampler [40] accessed through R version 3.6.1 [41] using the "runjags" package [42]. For the multi-site model, we used uninformative Uniform(lower = 0, upper = 1) priors for mean capture probability ($p$), weakly regularizing Normal(mean = 0, SD = 3.16)

priors for covariate effects on capture probability and half-Cauchy(scale = 1) priors for the standard deviation of temporal and site random effects in the CMR model (S1 Table). Priors for Mindego and San Bruno are presented in S1 Table. We ran the CMR model on five independent chains for 200,000 sampling iterations each after discarding the first 10,000 iterations as burn-in, and thinned the samples by a factor of 10, resulting in a final posterior sample of 100,000. For $N_a$ estimates, we report the mode of the posterior distribution followed by the 95% Highest Posterior Density Interval (HPDI). We also estimated the distribution of males versus females at each of the focal sites, as recovery criteria include establishing populations with equal sex ratios, and sex ratio bias is known to influence $N_e/N$ ratio estimates. To test for the correlation between the abundance ($N_a$) estimates and sex ratio bias, we performed a regression of modal values of $N_a$ and sex ratio for each site.

## Laboratory methods and bioinformatics

Genomic DNA was extracted using Qiagen DNeasy Blood and Tissue Kits (Qiagen, Valencia, California). Prior to next-generation sequencing (NGS) library preparation, we quantified DNA on a Qubit fluorometer (Life Technologies™), and 500 nanograms (ng) of DNA were used for library preparation. We followed the double-digest restriction-associated DNA (ddRAD) sequencing protocol developed in [43] for NGS library preparation, with some modifications. We digested genomic DNAs using 20 units each of the restriction enzymes *Sbf*I and *Msp*I (New England Biolabs, U.S.A.) and used Agencourt AMPure beads (Beckman Coulter, Danvers, Massachusetts) to purify the digestions prior to ligating uniquely bar-coded adapters with T4 ligase (New England Biolabs). We quantified all ligation products on the Qubit fluorometer, pooled across 12 index groups in equimolar concentrations, and then size selected fragments between 400 and 530 base pairs (bp) using a Pippin Prep size fractionator (Sage Science, Beverly, Mass.). We amplified the recovered fragments from each pool using 5–12 ng of the recovered DNA, Phusion High-Fidelty *Taq* (New England Biolabs), and Illumina's primers (Illumina, Inc., San Diego, California). Polymerase chain reaction (PCR) products were then cleaned with Agencourt AMpure beads (Beckman Coulter, Inc., Brea, California) and quantified using the Qubit fluorometer (Life Technologies) before being pooled for sequencing (100 bp single end reads) in a single lane on an Illumina HiSeq 4000 at the Genomics and Cell Characterization Core Facility at the University of Oregon.

We filtered and selected datasets using the STACKS version 2.3e [44] bioinformatics pipeline. We used the process_radtag program to clean and filter raw reads following default settings. We conducted initial parameter testing following the recommendations of [45]. This involved examining a series of *de novo* RAD locus assemblies that used a range of values for the mismatch distance between loci within an individual (*M*), the number of mismatches between loci in the catalogue (*n*) from 1 to 6 (fixing *n* = *M*), and the minimum stack depth (*m* = 2–4). The final set of parameters chosen for analysis was based on the total number of polymorphic loci shared by 80% of samples and how the distribution of SNPs per locus was affected. Once optimal parameters were selected (*m* = 3, *M* = 2, *n* = 3), we then executed the ustacks, cstacks, sstacks, and gstacks modules using the *denovo map* wrapper and generated four different datasets: (1) a dataset comprised of the seven focal sites from the years 2016–2018, hereafter called the 2018 dataset; (2) the temporal dataset, comprised of individuals collected from four sites (Pacifica, San Bruno, Pescadero, and Año Nuevo) at two different time periods (2004–2010 and 2016–2018) and used to assess change in genetic diversity over time and temporal estimates of effective population size (*Ne*); (3) a dataset comprised of the seven sites as in the 2018 dataset with five additional satellite sites from throughout the range, hereafter called the range-wide dataset; and (4) a phylogenetic dataset comprised of a subset of samples from all focal

and satellite sites, in addition to other subspecies of *T. sirtalis*. All datasets produced by STACKS were subjected to a final filter approach that retained loci present across all sampled sites with at least 80 percent (population structure and genetic diversity) and 60 percent (phylogenetic analyses) of the individuals sequenced.

## Population structure and genetic differentiation analyses

We evaluated population genetic structure with multiple analytical methods. First, we used the Bayesian clustering framework implemented in STRUCTURE version 2.3.4 [46] to identify genetic groups and estimate admixture levels among them. STRUCTURE uses a Markov Chain Monte Carlo (MCMC) method to simultaneously estimate population-level allele frequencies and group individuals into genetic clusters ($K$) that maximizes the within-cluster Hardy-Weinberg and linkage equilibria. Because there is often a large range of uncertainty in estimating $K$ [46–47] we used a hierarchical approach to identify the most probable number of clusters. First, we used the $\Delta K$ criterion to identify the highest hierarchical level of population structure [48]. Next, we tested for within-group structure by performing subsequent STRUCTURE analyses on each of the highest hierarchical clusters that were previously identified. If individuals could not be unambiguously assigned to a cluster (i.e. membership coefficients were < 0.60 for any cluster) we removed them from the dataset prior to the hierarchical STRUCTURE analyses. We also compared hierarchical $\Delta K$ results to the general (non-hierarchical) structure approach that uses log-likelihood values associated with each $K$ [46] and considered the spatial distribution of the sampled sites in relation to various cluster assignments. For all analyses, we used the rangewide dataset and the admixture model with correlated allele frequencies and estimated the probability of $K$ (1–15) clusters. For each $K$ that was evaluated, we used 10 separate runs with 500,000 iterations of the MCMC algorithm following a burn-in of 500,000 iterations and calculated the mean log probability of the data ($\ln\Pr(X|K)$ in [46]) for the 10 runs combined. Results were compiled graphically in CLUMPAK [49].

To complement the STRUCTURE analyses, we used discriminant analysis of principal components (DAPC), a multivariate ordination approach implemented in the R package ADEGENET version 2.1.0 [50]. DAPC is a non-model based method and does not require the assumption of HWE or unlinked markers, as in the STRUCTURE analyses. This method evaluates the optimal number of genetic clusters using PCA ordination to maximize the between-group variation while minimizing the variation found within groups. Given that DAPC relies on data transformation using PCA as a prior step, retaining too many PCs can lead to overfitting the discriminant functions. We used the cross-validation function *xvalDapc* in ADEGENET to identify the optimal number of PCs to retain. The DAPC procedures were replicated 100 times at each level of PC retention, and we selected the number of PCs associated with the lowest root mean squared error (RMSE) for the final analysis.

When a species' distribution is characterized by isolation by distance (IBD), where genetic differentiation among populations is correlated with the geographic distance that separates them, it can be difficult for clustering methods to distinguish between genuine genetic clusters and artefacts due to IBD [47, 51]. To evaluate this relationship, we plotted pairwise estimates of differentiation by geographic distance using the rangewide dataset and used Mantel tests to assess correlation between pairwise genetic and geographic distances [52]. The magnitude of population differentiation ($F_{ST}$) was estimated using Weir & Cockerham's θ [53], an unbiased estimator of $F_{ST}$. To visualize genetic distances with geography, individual point locations and pairwise genetic distances among individuals were analyzed with the package MAPI v. 1.0.0–62 [54] in R 3.6.1. The analysis highlights spatially connected cells with genetic distance values that are significantly high (genetically discontinuous) or low (genetically continuous) in

comparison to random permutations of values among sample locations, which provides a graphical representation of genetic distances among samples without the confounding effects of IBD [53]. We calculated codominant genotypic distances [55–56] among all individuals, georeferenced to their collection locations. Ellipses representing the pairwise genetic distances among points were overlaid on a grid, and grid cell values were computed as the average of overlapping ellipses. Cell sizes were computed in MAPI as a function of the study area and number of samples, using a beta = 0.25 for random (versus regular) sampling [57] and significance determined with 1000 permutations. We exported the results as ESRI shapefiles and visualized them in ArcMap 10.4.1. We also assessed whether selection may contribute to population structure across the San Francisco Peninsula by using the $F_{ST}$-outlier approach implemented in BAYESCAN v2.1 [58]. This Bayesian method identifies putative outlier loci under selection because they show $F_{ST}$ coefficients that are significantly more different than expected under neutrality. BAYESCAN analyses consisted of 20 pilot runs of 5,000 iterations with a burn-in of 50,000 iterations. We used a thinning interval of 10 for a total number of 100,000 iterations, and ran two analyses, one with the prior odds for the neutral model set at 100 and the other analysis with the prior odds set at 500. To assess the contribution of neutral SNPs versus SNPs potentially under selection in delineating population structure, we repeated the STRUCTURE analysis and pairwise estimates of differentiation (θ) using only the SNPs identified as outliers.

Finally, we used MRBAYES 3.2 [59] to estimate a phylogenetic tree of genetic relationships between sites and genetic clusters. We randomly selected three samples per site from the rangewide *T. s. tetrataenia* dataset, with the exception of Site 5 where only two samples were collected. We also included a single sample of the red-sided gartersnake (*Thamnophis sirtalis infernalis*) from Lake Lagunita, Stanford University, Santa Clara County, California, USA (CAS 201525), and a single sample of the valley gartersnake (*Thamnophis sirtalis fitchi*; CAS 212588) from Colusa County, California, USA. A single sample of the undescribed south coast gartersnake (*Thamnophis sirtalis* ssp.), taken from the Santa Margarita River, San Diego County, California, USA, was used to root the tree because this sample represents the furthest geographic distance away from the San Francisco Peninsula and it is currently treated as an undescribed taxon [60–62]. Tree searches consisted of two independent MCMC searches of tree space for 5 million generations each, sampling every 1,000 steps and discarded the initial 25% of samples from each run. We assessed evidence for convergence using tracer 1.7.1 [63] and considered lineages with posterior probabilities ≥ 0.95 to be strongly supported.

## Effective size estimation and genetic rescue

We calculated summary statistics in STACKS to compare genetic diversity among sites and genetic clusters with the 2018 and temporal datasets. Summary statistics include the following: allelic richness (*Ar*), mean observed heterozygosity (*Ho*), mean expected heterozygosity (*He*), and mean nucleotide diversity (*π*). We estimated the contemporary effective size ($N_e$) of each site with the 2018 dataset and temporal dataset. We relied on the $N_e$ thresholds outlined in Frankham et al. [64] to help guide management considerations. They conclude that a minimum $N_e \geq$ 100 as a short-term goal avoids the risk of extinction owing to inbreeding depression. We used the linkage disequilibrium method (LDNe) [65] and when possible a two-sample temporal method using moment-based F-statistics [66] within the program NEESTIMATOR v2.1 [67] to obtain $N_e$ values. The LDNe method has been shown to generally outperform coalescent-based methods for $N_e$ estimation [68]. We assumed random mating at each site, calculated 95% confidence intervals for point estimates using the jackknife-across-samples method [69], and screened out rare alleles using a critical cut-off value (*Pcrit*) of 0.05. We used $N_e$ values and the adult population size estimates ($N_a$) to compute the $N_e/N_a$ ratio for each site for each year.

We evaluated whether the seven focal sites sampled in our 2018 dataset met criteria that would indicate that genetic erosion has occurred and whether genetic rescue could increase heterozygosity. Following the equation of Frankham et al. [13], we calculated the mean inbreeding coefficient ($F$) for each site as the ratio of average heterozygosity ($H$) of the receiver site (inbred) to the proposed heterozygosity of the source site (outbred). We used the suggested threshold of $F > 0.01$ [13] to identify receiver populations with genetic erosion in comparison to the source population that could benefit from assisted migration and augmentation management. To capture potential inbreeding effects of both genetic drift and non-random mating, we used observed heterozygosity estimates for each receiver site and expected heterozygosity estimates for the source site [13]. We split source-recipient comparisons according to the regional clusters identified from populations structure analyses and computed the mean inbreeding coefficient ($F$) for each site in relation to the following source population scenarios: (i) the closest source site to the receiver site, (ii) the source site with largest $N_e$, (iii) the source site with the highest $He$, and (iv) the source site with largest adult population size ($N_a$).

## Results

### Population size and sex ratio estimates

We made 1088 captures of 815 individual *T. s. tetrataenia*, over 34,576 trap-nights of sampling at the seven sites from 2016–2018 (Table 1). We captured 362 females, 392 males, and 7 snakes of unknown sex. The highest abundance estimate ($N_a$) of *T. s. tetrataenia* was at San Bruno (modal $\hat{N}_a = 1317$, 95% HPDI = 1145–1487), followed by Mindego ($\hat{N}_a = 204$, 95% HPDI = 144–328), Año Nuevo ($\hat{N}_a = 123$, 95% HPDI = 93–161), and Pescadero ($\hat{N}_a = 73$, 95% HPDI = 61–88) (Fig 2). Abundance estimates were nearly equal at Pacifica ($\hat{N}_a = 47$, 95% HPDI = 33–63), Skyline ($\hat{N}_a = 48$, 95% HPDI = 33–61), and Crystal Springs ($\hat{N}_a = 50$, 95% HPDI = 35–64). In general, posterior distributions of sex ratios (males/females) were more female-biased in northern regional sites and male-biased in southern regional sites (Fig 3). Regression analysis of $N_a$ estimates and sex ratios were not significant when all sites were included ($R^2 = -0.19$, $p = 0.99$). However, this was primarily due to the large $N_a$ estimate at San Bruno relative to all other sites. If San Bruno was treated as an outlier site and excluded from the analysis, then a significant correlation between $N_a$ and sex ratio bias was observed ($R^2 = 0.55$, $p = 0.05$). However, the sex ratio did not significantly differ from 1 at any site, on the basis of whether the 95% HPDI overlapped 1.

### Summary of bioinformatics for genetic analyses

Using the stacks pipeline, our ddRAD sequencing effort yielded an average of 3,368,646 sequences per-individual (median: 3,276,428; min: 94,562; max: 8,250,652) across the 248 individuals sequenced. The mean coverage depth per-individual was 61.4X (min: 27.3X; max: 94.7X). After merging and calling final consensus sequences, we obtained 121,639 loci across 186 individuals sequenced. Once final filters were applied, we obtained 3,788 SNPs for the 2018 dataset, 2,747 SNPs for the temporal dataset, 3,029 SNPs in the rangewide dataset, and 7,036 SNPs in the phylogenetic dataset.

### Population structure and genetic differentiation

Using a hierarchical approach and the ΔK statistic, STRUCTURE partitioned *T. s. tetrataenia* sites into two regional clusters (Fig 4): (i) a "northern" regional cluster that extends from Pacifica and San Bruno southward along the San Andreas rift valley (Skyline, Site 1 & 2, and Crystal

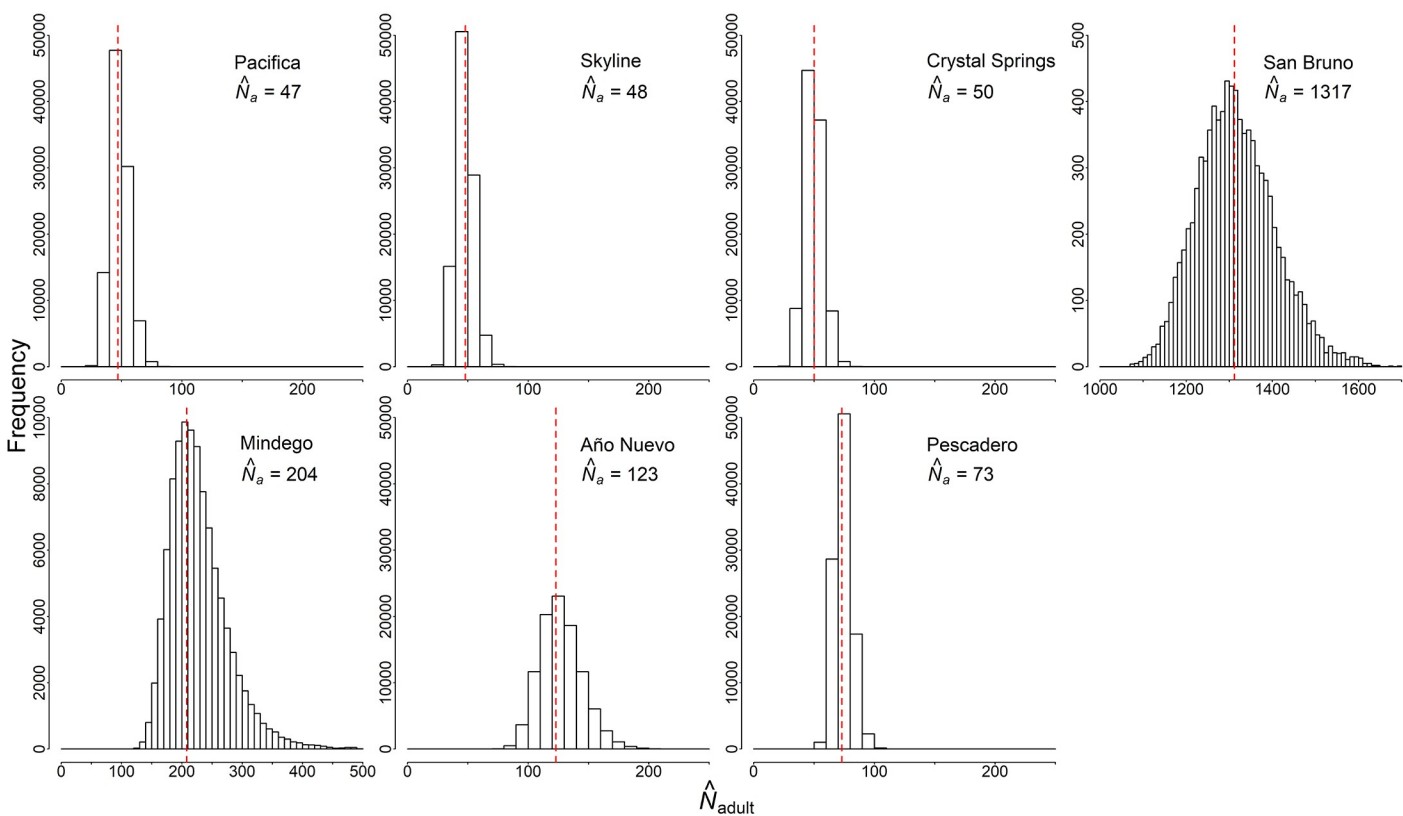

**Fig 2. Posterior distributions of estimated adult abundance ($N_a$) at seven sites sampled for San Francisco gartersnakes (*Thamnophis sirtalis tetrataenia*) in 2016 (Mindego), 2017 (San Bruno), and 2018 (Año Nuevo, Pescadero, Pacifica, Skyline, and Crystal Springs).** The posterior distribution is displayed as a histogram of the frequency of possible $N_a$ values for each site. Red dashed lines and $\hat{N}_a$ values presented in each panel represent the mode of the posterior distribution for adult abundance at that site.

Springs), and (ii) a "southern" regional cluster that extends from Mindego westward to the coastal sites of Site 4, Pescadero, Año Nuevo, and Site 5. Membership coefficients ($Q$) of individuals from Site 3 showed nearly equal proportions to both northern and southern regional clusters ($Q = 0.567/0.433$, respectively) and were admixed between clusters up to $K = 8$, where they assigned to a distinct genetic cluster. Therefore, this site was excluded from the dataset before performing within-group analyses of regional clusters. Within the northern regional cluster, four miniclusters were supported (Fig 4): (i) Pacifica, (ii) Skyline, (iii) Sites 1 and 2 and Crystal Springs, and (iv) San Bruno. Snakes along the San Andreas rift valley showed some levels of admixture with each other, whereas assignment proportions for snakes from both Pacifica and San Bruno were exclusive. Similarly, within-cluster analyses across the southern regional cluster supported three additional miniclusters ($\Delta K = 3$): (i) Mindego, (ii) Pescadero, Sites 4 and 5, and (iii) Año Nuevo. Some individuals sampled at Año Nuevo were admixed with the minicluster containing Pescadero, Site 4 and 5. The results of each $K$ evaluated ($K = 1$–$8$) are presented in S2 Fig.

The DAPC cross-validation indicated 30 PCs provided the highest mean success of assignment at 0.953 with a lowest RMSE of 0.062. The final DAPC analysis resulted in seven discriminant functions, with a 0.536 proportion of conserved variance. The resulting DAPC plots largely discriminated the same seven miniclusters as the STRUCTURE within-group analyses (Fig 5). The first two discriminant functions strongly discriminated among Pacifica, San Bruno

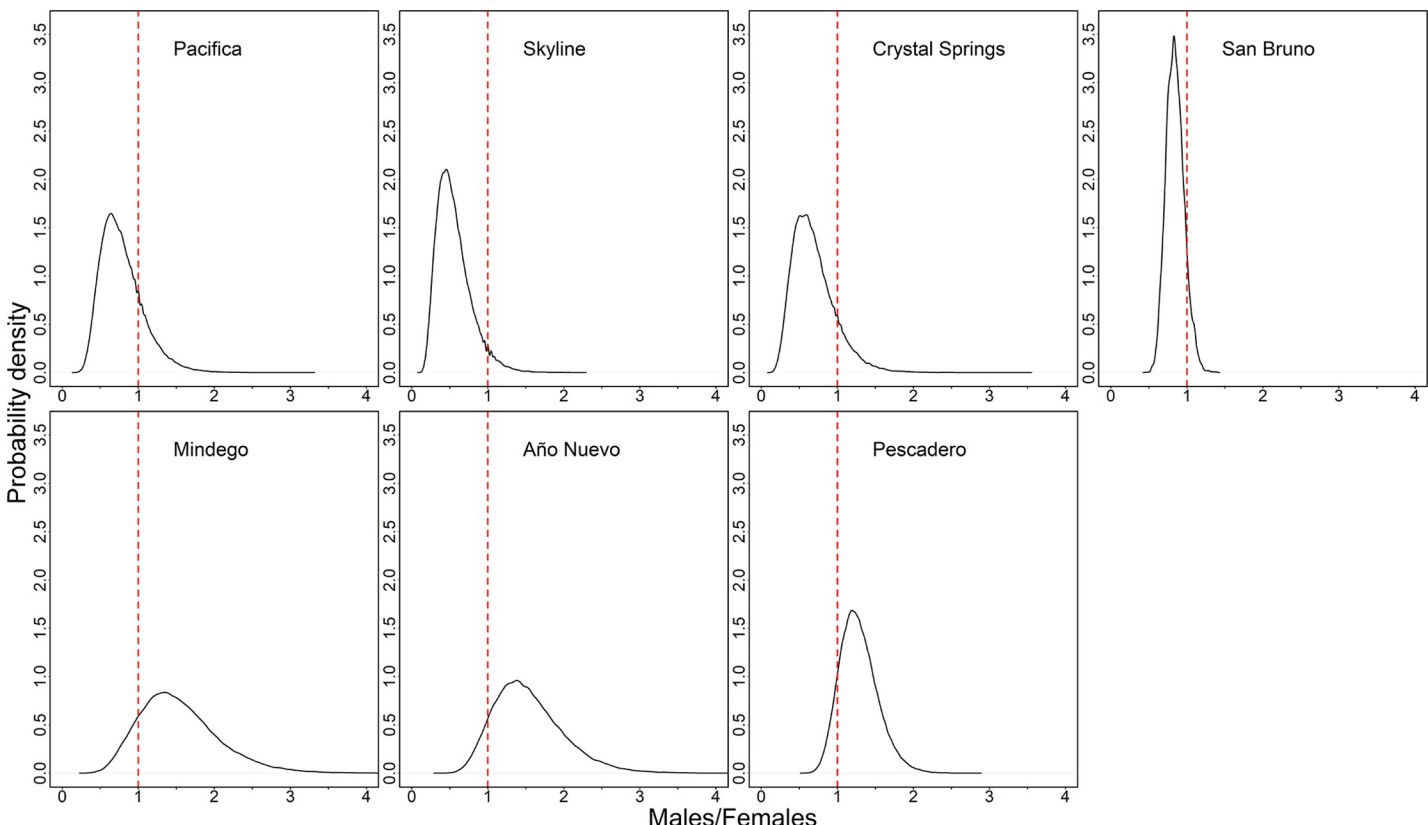

**Fig 3. Posterior distributions of estimated adult sex ratio (males/females) at seven sites sampled for San Francisco gartersnakes (*Thamnophis sirtalis tetrataenia*) in 2016 (Mindego), 2017 (San Bruno), and 2018 (Año Nuevo, Pescadero, Pacifica, Skyline, and Crystal Springs).** The posterior distribution is displayed as the probability density of different sex ratio values for each site. Red dashed lines in each panel represent a 1:1 sex ratio. Values to the left of the red dashed line represent a sex ratio biased towards females, values to the right of the red dashed line represent a sex ratio biased towards males. If the posterior distribution broadly overlaps both sides of the red dashed line, there is no evidence of a biased sex ratio in that population.

and the San Andreas rift valley in the north (Fig 5A); however, there was minimal discrimination among sites within the San Andreas rift valley. The DAPC plot using the third and fourth discriminant functions showed Mindego separated from most other sites in the south (Fig 5B), while all other sites occupied distinct but similar linear space.

BAYESCAN analyses (run with prior odds set at 100 and 500) identified 12 outlier SNPs with $F_{ST}$ coefficients that were significantly more different than expected under neutrality. Population structure analyses using these 12 loci did not produce any meaningful patterns (S3 Fig) indicating that there was little effect of selection in our dataset. We detected a significant pattern of increasing genetic isolation with geographic distance among sites using the rangewide dataset (Fig 6A; $R^2$ = 0.17, p = 0.008). The inter-individual genotypic distances mapped using the software MAPI identified two areas associated with significantly higher genetic dissimilarity (Fig 6B). The first area of higher genetic discontinuity is broadly centered along the Santa Cruz Mountains and separates the north and south regions, similar to the cluster analyses. In northern San Mateo County, cells separating San Bruno from other sites received "hotter' values of genetic discontinuity, highlighting the isolation of individuals at this site. Global $F_{st}$ among the sites sampled based on the rangewide dataset was 0.148, $F_{st}$ between the northern and southern regional clusters was 0.077, and pairwise $F_{st}$ estimates between sites with $\geq$ 5 samples were significant and ranged from 0.039 to 0.218 (Table 2). Temporal pairwise estimates of genetic differentiation also increased between the 2004–2010 and 2016–2018 sample

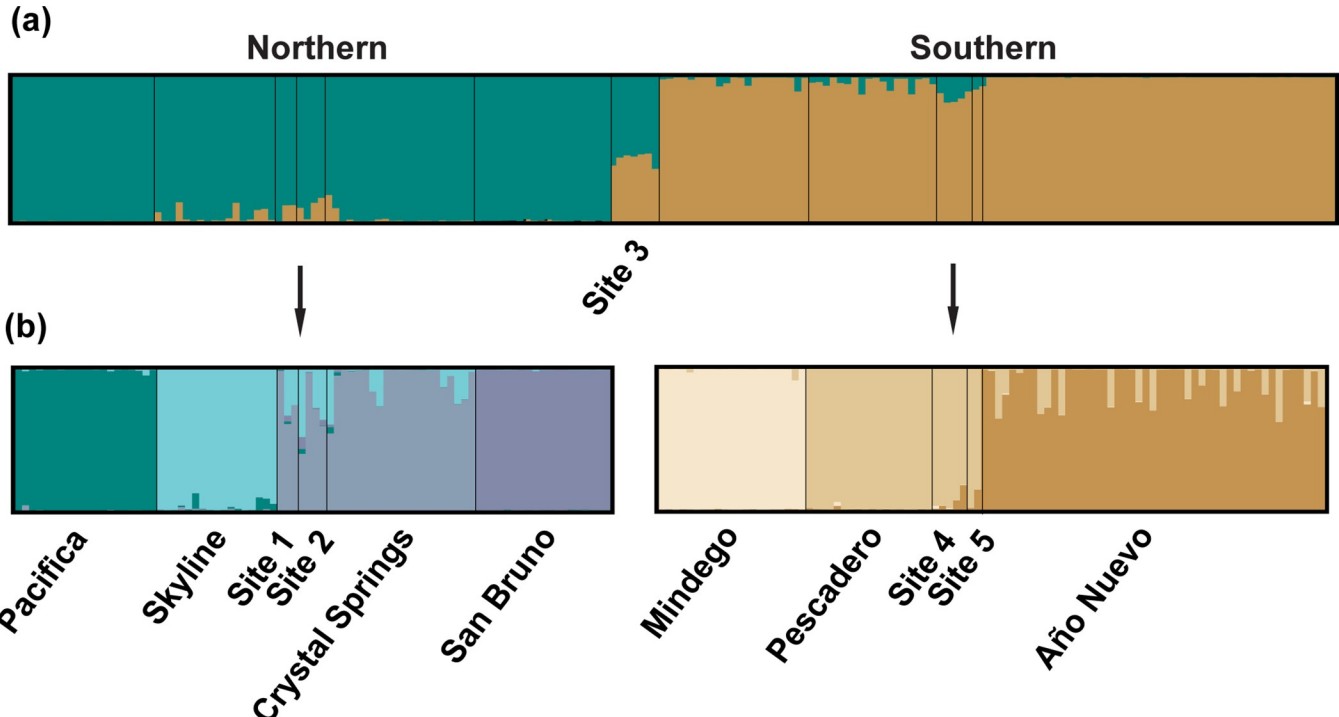

**Fig 4. Cluster assignments of individuals from the sites sampled across the range of *T. s. tetrataenia* sites.** STRUCTURE, assignments (a) show the two clusters identified at K = 2 that separate sites into northern and southern regional clusters, and (b) hierarchical analysis within each regional cluster that supported four miniclusters (northern) and three miniclusters (southern).

periods (Fig 7), with site comparisons involving Pacifica and San Bruno showing the largest changes.

Phylogenetic tree estimation recovered two regional clades corresponding to northern and southern *T. s. tetrataenia* that are consistent with cluster analyses (Fig 8). We find strong support for relationships within the Northern clade. Pacifica and Skyline form distinct groups that are sister to a group corresponding to the remaining San Andreas rift valley sites, and San Bruno forms a distinct group that is sister to all other sites nested in the Northern clade. With the exception of Pescadero, sites included in the southern clade tended to form distinct groups, but the relationships among sites were not well supported. A third clade, comprised of different *T. sirtalis* subspecies, renders the two *T. s. tetrataenia* clades paraphyletic. This clade is sister to the northern clade and includes individuals from two sites that are often treated as *T. s. tetrataenia-T.s.infernalis* intergrades (Site 3 and Lake Lagunita, Stanford University) as well as a single sample of *T. s. fitchi* from Colusa County.

### Genetic diversity, effective size, $N_e/N_a$ ratios, and genetic rescue options

Diversity estimates were generally similar across sites, with the exception of Pacifica where measurably lower values were observed (Table 3). The highest estimates of diversity were found at Crystal Springs and Pescadero. Regional cluster diversity estimates were highest in the southern cluster for all diversity indices except allelic richness. Temporal estimates of diversity decreased between the 2004–2010 and 2016–2018 samples at Pacifica and Pescadero, while San Bruno and Año Nuevo increased in genetic diversity (Table 3). Effective population size ($N_e$) estimates were below the short-term threshold recommendation to limit inbreeding depression ($\geq$ 100) for most sites with point estimates ranging from 9 to 60 (Table 4). The

**(a)**

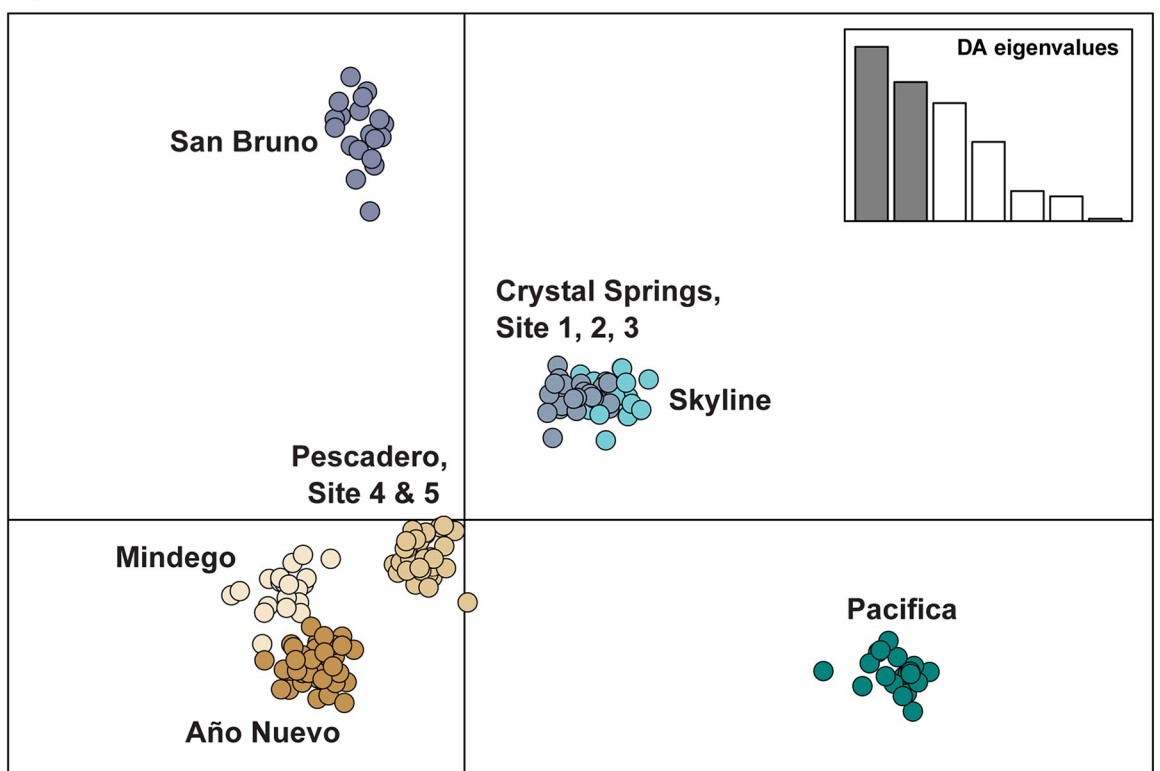

**(b)**

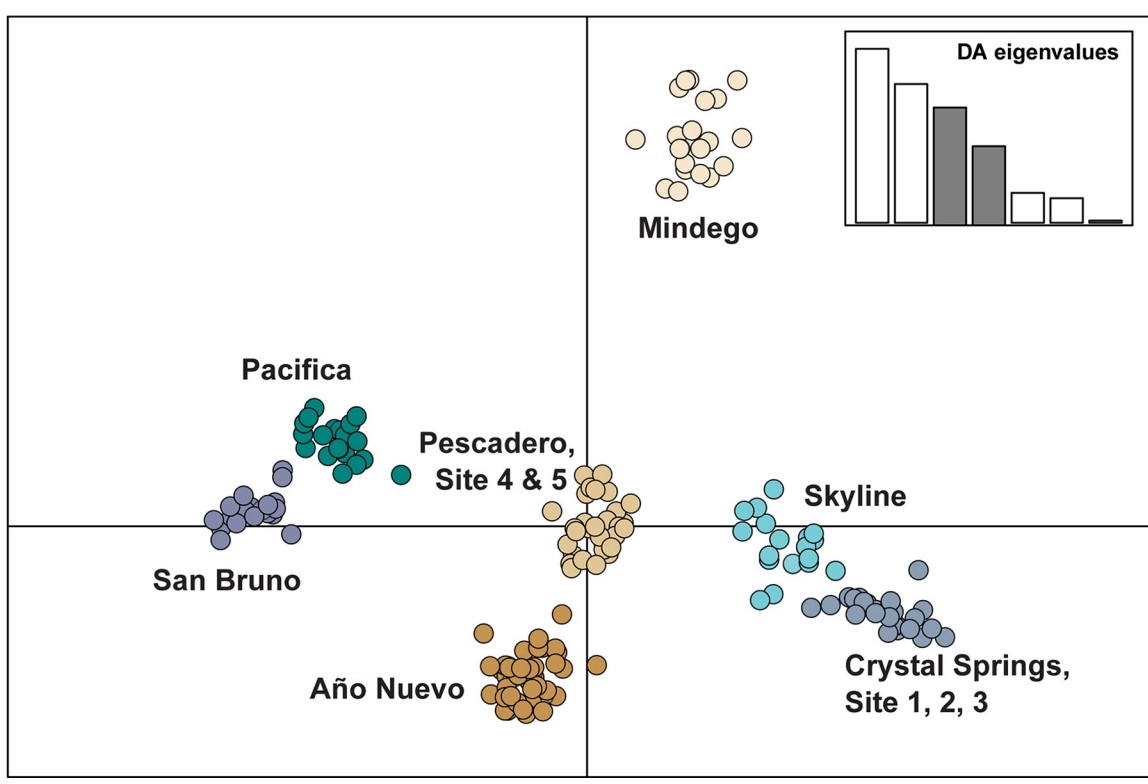

**Fig 5.** DAPC scatterplots of sampled individuals using (a) discriminant functions 1 and 2, and (b) discriminant functions 3 and 4 (discriminant functions are indicated by the grey bars).

only exception was San Bruno, where $N_e$ was estimated at 254. Effective population size estimates were particularly low at Pacifica and Crystal Springs ($N_e$ = 9–13 and $N_e$ = 10, respectively). Estimates using the two-sample temporal method provided similar point values and smaller confidence intervals than the LDNe method (Table 4). Using the point estimates for $N_e$ and $N_a$, we estimated an $N_e/N_a$ ratio for each site. The highest ratio was at Pescadero ($N_e/N_a$ = 0.78), followed by Skyline ($N_e/N_a$ = 0.64). The remainder of the sites were lower and ranged from 0.16–0.34.

We used estimates of inbreeding coefficients (*F*) to investigate evidence of genetic erosion. Within northern sites, we estimated mean inbreeding coefficients (*F*) greater than 10 percent at Pacifica across all four genetic rescue scenarios (Table 5). This population also met other criteria that suggest genetic erosion (small, $N_a$ & $N_e$ < 100 and isolated). For San Bruno, the mean inbreeding coefficient (*F*) was high using the nearest source site (Skyline) and the source site with highest heterozygosity (Crystal Springs), but *F* did not exceed the 10% threshold and

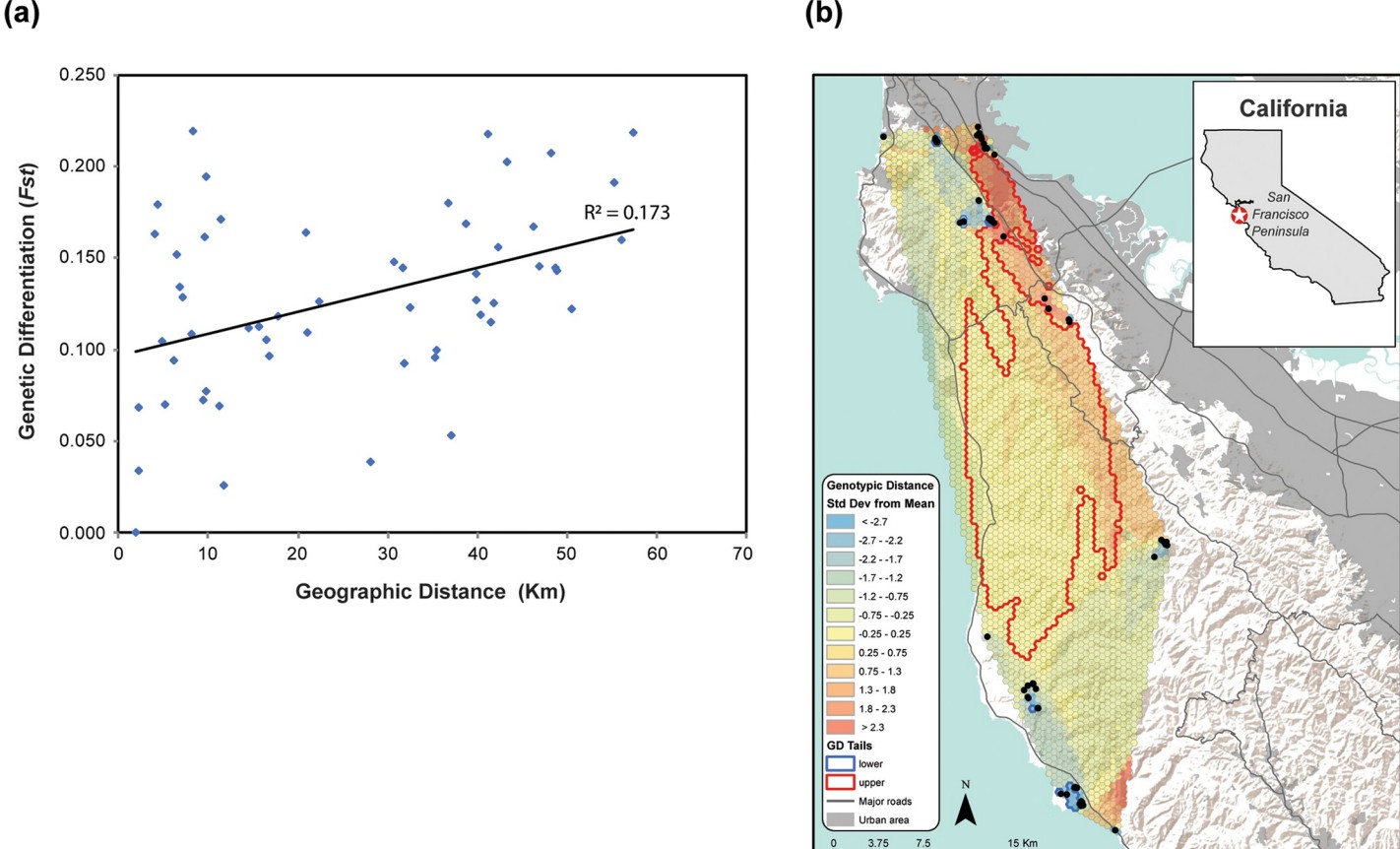

**(a)**

**(b)**

**Fig 6. Isolation by distance of pairwise genetic differentiation (θ) estimates and MAPI inter-individual pairwise distance surface using genotypic distances.** Mantel tests confirmed a significant positive correlation ($R^2$ = 0.173; p = 0.008). World Terrain Basemap source: USGS, ESRI, NOAA. Urban land coverage for San Francisco Bay Region source: Bay Area Open Space Council, GreenInfo Network, Conservation Lands Network, and San Francisco Bay Area Upland Habitat Goals Project. (2011). California Urban Lands: Farmland Mapping and Monitoring Project, 2006. Bay Area Open Space Council. Available at http://purl.stanford.edu/kh450fm7856.

**Table 2. *Thamnophis sirtalis tetrataenia* pairwise genetic differentiation estimates (θ; Weir & Cockerham [51]) for sampled sites with > 5 samples/site.**

| Sites | Pacifica | Skyline | Crystal Springs | San Bruno | Site 3 | Mindego | Site 4 | Pescadero |
|---|---|---|---|---|---|---|---|---|
| **Skyline** | 0.179 | - | | | | | | |
| **Crystal Springs** | 0.171 | 0.109 | - | | | | | |
| **San Bruno** | 0.219 | 0.163 | 0.152 | - | | | | |
| **Site 3** | 0.164 | 0.118 | 0.077 | 0.113 | - | | | |
| **Mindego** | 0.217 | 0.168 | 0.148 | 0.180 | 0.109 | - | | |
| **Site 4** | 0.207 | 0.146 | 0.127 | 0.167 | 0.092 | 0.097 | - | |
| **Pescadero** | 0.203 | 0.156 | 0.100 | 0.125 | 0.039 | 0.106 | 0.070 | - |
| **Año Nuevo** | 0.218 | 0.159 | 0.145 | 0.191 | 0.119 | 0.126 | 0.112 | 0.072 |

Statistical significance at α < 0.002 after Bonferroni correction, all values were significant.

San Bruno did not meet other genetic erosion criteria except being isolated. Within the southern sites, only Mindego showed evidence of genetic erosion. Estimates of *F* for Mindego were marginally greater than 10% using Pescadero as the source site and this site is geographically

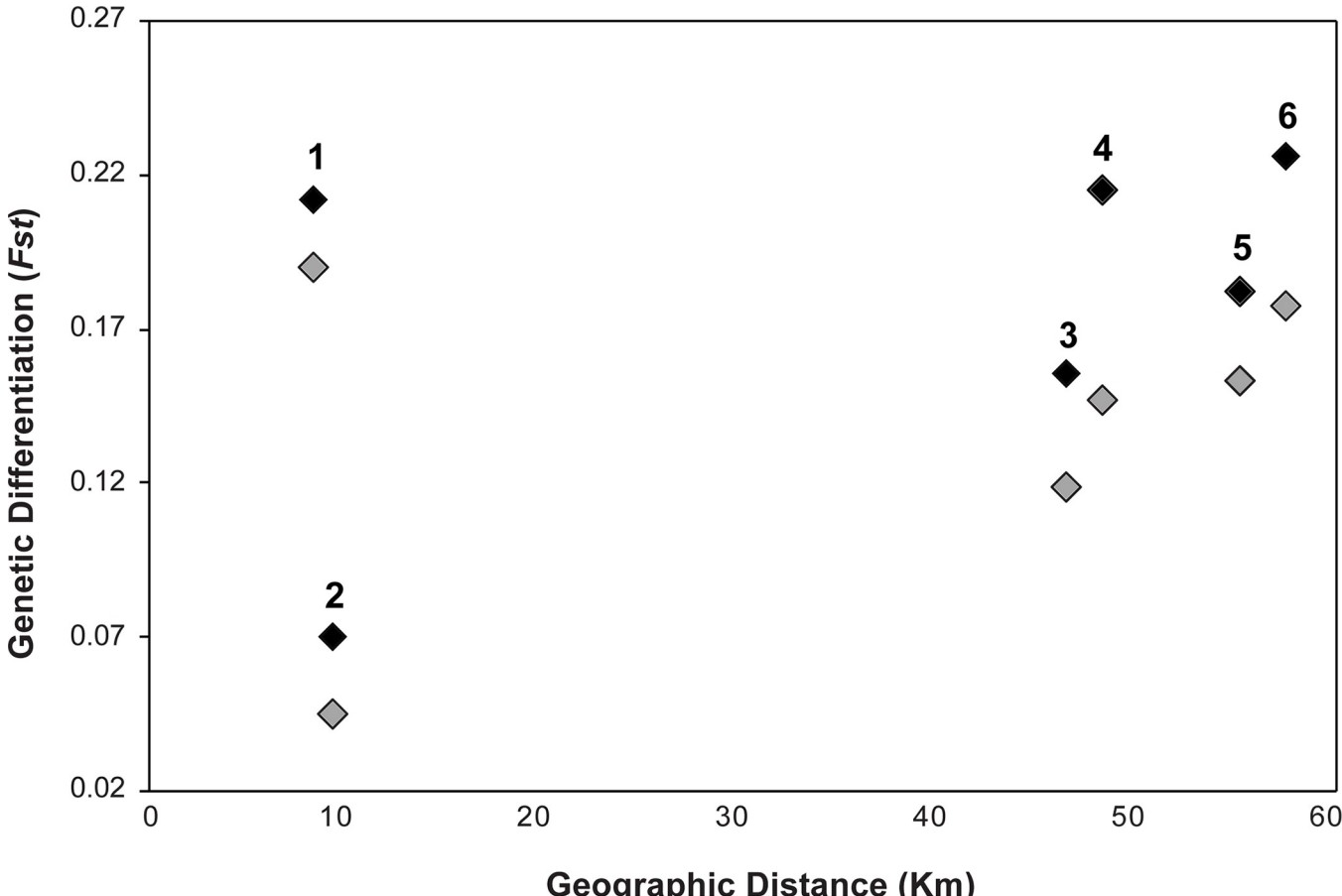

**Fig 7. Pairwise genetic differentiation estimates plotted by geographic distance among four sites (Pacifica, San Bruno, Pescadero, and Año Nuevo) in kilometers.** Light grey diamonds are pairwise estimates between sites measured in 2005, and black diamonds from 2018: (1) Pacifica vs. San Bruno, (2) Pescadero vs. Año Nuevo, (3) San Bruno vs. Pescadero, (4) Pacifica vs. Pescadero, (5) San Bruno vs. Año Nuevo, and (6) Pacifica vs. Año Nuevo. All values increased over time.

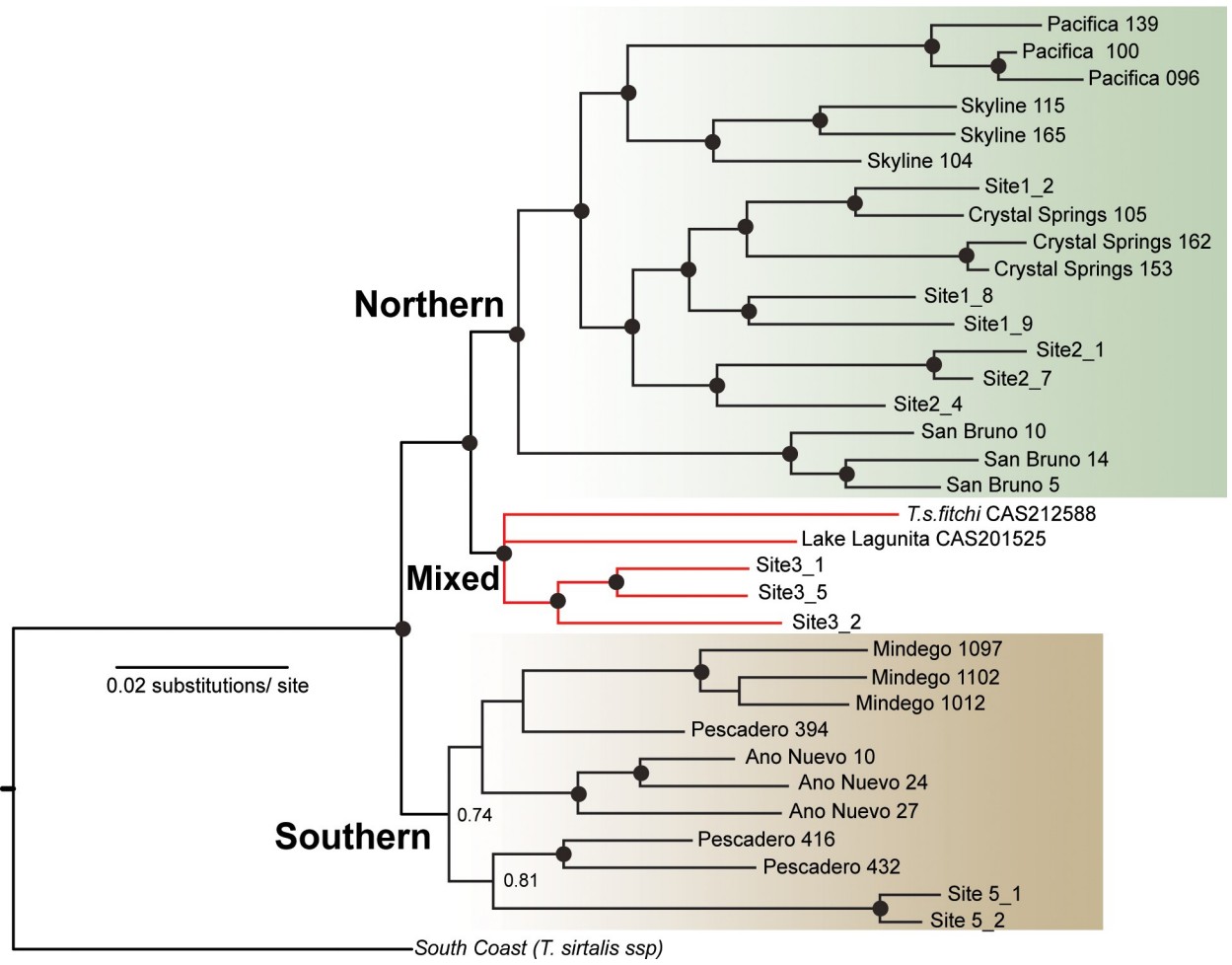

**Fig 8. Phylogenetic tree of *Thamnophis sirtalis tetrataenia* based on 7,036 loci using MrBayes.** The northern and southern clades within *T. s. tetrataenia* are colored according to population cluster analyses, and samples are labeled by sites sampled. Black dots indicate branch support of ≥ 0.95 posterior probability.

isolated with a low effective size ($N_e < 100$). Overall, these results suggest that several sites may be suffering from genetic erosion.

## Discussion

Using a combination of genomic and demographic data, we show evidence of regional partitioning and differences in the amount of genetic diversity across the range of an endangered vertebrate within a landscape that includes both urban and natural barriers. Genetic partitioning is consistent with the effects of distance in a movement-limited organism and degree of isolation, as well as local demographic artifacts that drive differentiation in the absence of gene flow. Below we detail these findings and discuss their implications for management.

### Regional population structure

Phylogenetic, clustering, and genetic differentiation analyses all supported two regional groups of *T. s. tetrataenia*, a northern and southern group. While connectivity throughout most of the San Francisco Peninsula has waned over the last century due to urban expansion (Fig 1), long standing environmental gradients along the Santa Cruz Mountains may have also presented

**Table 3. Diversity statistics based on SNPs from the "2018 dataset" and "Temporal dataset".**

| Sites (2018) | Year | N | Ar | *H*o | *H*e | *π* |
|---|---|---|---|---|---|---|
| **Pacifica** | 2018 | 20 | 1.31 | 0.092 | 0.090 | 0.092 |
| **Skyline** | 2018 | 17 | 1.41 | 0.123 | 0.123 | 0.127 |
| **Crystal Springs** | 2018 | 17 | 1.44 | 0.127 | 0.123 | 0.127 |
| **San Bruno** | 2017 | 20 | 1.42 | 0.114 | 0.114 | 0.117 |
| **Mindego** | 2016 | 22 | 1.38 | 0.112 | 0.112 | 0.115 |
| **Pescadero** | 2016 | 18 | 1.45 | 0.122 | 0.124 | 0.128 |
| **Año Nuevo** | 2018 | 49 | 1.41 | 0.120 | 0.121 | 0.122 |
| *Northern* | 2016–2018 | 74 | 1.76 | 0.110 | 0.117 | 0.118 |
| *Southern* | 2016–2018 | 89 | 1.62 | 0.116 | 0.122 | 0.122 |
| **Sites (Temporal)** | **Year** | **N** | **Ar** | ***H*o** | ***H*e** | ***π*** |
| **Pacifica** | 2004–2006 | 13 | 1.36 | 0.110 | 0.105 | 0.107 |
| | 2018 | 19 | 1.29 | 0.099 | 0.096 | 0.098 |
| **San Bruno** | 2006–2007 | 13 | 1.39 | 0.112 | 0.113 | 0.118 |
| | 2017 | 20 | 1.40 | 0.117 | 0.118 | 0.121 |
| **Pescadero** | 2005–2010 | 18 | 1.50 | 0.132 | 0.134 | 0.139 |
| | 2016 | 18 | 1.46 | 0.130 | 0.132 | 0.136 |
| **Año Nuevo** | 2005–2006 | 18 | 1.41 | 0.123 | 0.121 | 0.125 |
| | 2018 | 49 | 1.41 | 0.127 | 0.127 | 0.129 |

Sites = each sampled site, N = mean number of individuals per locus, Ar = allelic richness, Ho = mean observed heterozygosity, He = mean expected heterozygosity, $π$ = mean nucleotide diversity. Allelic richness (Ar) estimates were rarified by lowest number of gene copies per site/cluster (2018 dataset: 28/122; Temporal dataset: 22).

movement barriers for *T. s. tetrataenia* even prior to urbanization. For example, freshwater wetlands with heterogeneous grassland-scrub upland communities are key habitat components that support perennial *T. s. tetrataenia* populations and their co-occurring amphibian prey species [70–73], but they predominantly occur at lower elevations along the east and west sides of the Santa Cruz Mountains. In contrast, higher elevations along the crest of the mountains are dominated by the cooler, redwood-douglas fir forests [74–75]. Where wetland habitat is available at high elevation, such as Mindego (~550m), it tends to be naturally isolated from other such sites (Fig 1). The upland grassland-scrub communities at Mindego are bounded by the mixed evergreen forests that separate this population by several kilometers from other permanent wetlands in the region [72]. Although isolated, our genetic data demonstrate that this population is more closely allied to the southern regional group compared to the northern group, suggesting that a reduction in gene flow in the middle of the species range is due to a combination of habitat limitations and geographic isolation. Further sampling in geographically intermediate areas would provide a better assessment of genetic interactions between the two major groups, but nonetheless these data support the recognition of these historically distinctive groups as separate management units.

Our results also corroborate past findings of a putative boundary between *T. s. tetrataenia* and *T. s. infernalis*, another subspecies of the common gartersnake (*T.sirtalis*) that borders the southern range of *T. s. tetrataenia* [76–77]. Studies of color pattern have found a high proportion of intermediate color patterns between the two subspecies along the east side of the Santa Cruz Mountains, from Site 3 to as far south as Palo Alto [76–77]. Our phylogenetic analyses grouped individuals sampled at Site 3 into a 'mixed clade' that included other samples of *T. sirtalis* and was sister to the northern *T. s. tetrataenia* clade (Fig 5). In light of both morphological and genetic patterns, the southern San Andreas rift valley, from the region of Site 3 southward,

**Table 4. $N_e$ estimates using single sample (LDNe) and two-sample temporal methods, year(s) sampled, and $N_e/N_a$ ratios across the seven focal sites.**

| Population | LDN$_e$ | | | Temporal $N_e$ | | | $N_e/N_a$ ratio |
|---|---|---|---|---|---|---|---|
| | $N_e$ | 95% CI | Year | $N_e$ | 95% CI | Years | |
| **Pacifica** | 13 | 8–24 | 2018 | 9 | 8–9 | 2004–2006 2018 | 0.20 |
| **Skyline** | 30 | 17–83 | 2018 | – | – | – | 0.64 |
| **Crystal Springs** | 10 | 5–23 | 2018 | – | – | – | 0.20 |
| **San Bruno** | 255 | 46 –INF | 2017 | 254 | 145–1021 | 2006–2007 2017 | 0.19 |
| **Mindego** | 33 | 17–139 | 2016 | – | – | – | 0.16 |
| **Pescadero** | 60 | 23 –INF | 2016 | 56 | 49–66 | 2005–2010 2016 | 0.78 |
| **Año Nuevo** | 41 | 30–60 | 2018 | 49 | 44–56 | 2005–2006 2018 | 0.40 |

Jackknife on loci was used to estimate upper and lower confidence intervals. INF indicates an estimated confidence interval of "infinity", suggesting there is not enough information to obtain a reliable estimate. A dash (-) indicates there were no samples available for that site/year. The critical value was set at 0.05 to screen out rare alleles. For $N_e$ /$N$ ratios, we used the temporal $N_e$ estimate when available otherwise the LDNe estimate was used.

likely represents a region of gene exchange between these recently diverged taxa. Previous investigations of relationships among *T. sirtalis* subspecies across western North America using mtDNA [78] and both mtDNA and five microsatellites [79] have provided little evidence of diversification among *T. sirtalis* subspecies in this region. Although outside the scope of our study, future investigations using extended sampling of *T. sirtalis* subspecies and genome-wide SNP data would provide greater number of markers and may help to better evaluate the extent of gene flow and taxonomic limits within this region.

## Population size and genetic diversity

Given the protected status of *T. s. tetrataenia* and the possibility of isolation among populations due to human alterations of the landscape, we evaluated effective population size ($N_e$)

**Table 5. Assessment of genetic erosion and the expected impact of genetic rescue scenarios on recipient populations using different donor sites and scenarios based on the mean inbreeding coefficient ($F$) calculated following the equation of Frankham et al. [13].**

**Northern Regional Populations**

| Recipient | Source sites | | | | Pop. Isolated? | Pop. small? | $F > 0.1$ |
|---|---|---|---|---|---|---|---|
| | Neighbor $F$ (varies*) | $N_e$ F (San Bruno) | He F (Crystal Springs) | $N_a$ F (San Bruno) | | | |
| **Pacifica** | 0.28 | 0.21 | 0.28 | 0.21 | Yes | Yes | Yes |
| **Skyline** | 0.03 | -0.05 | 0.03 | -0.05 | Yes | Yes | No |
| **Crystal Springs** | 0.00 | -0.09 | - | -0.09 | Yes | Yes | No |
| **San Bruno** | 0.10 | - | 0.10 | - | Yes | No | No |

**Southern Regional Populations**

| Recipient | Source sites | | | | Pop. Isolated? | Pop. small? | $F > 0.1$ |
|---|---|---|---|---|---|---|---|
| | Neighbor $F$ (varies*) | $N_e$ F (Pescadero) | He F (Pescadero) | $N_a$ F (Mindego) | | | |
| **Mindego** | 0.13 | 0.13 | 0.13 | - | Yes | No | Yes |
| **Pescadero** | 0.00 | - | 0.00 | -0.06 | No | No | No |
| **Año Nuevo** | 0.06 | 0.06 | - | -0.04 | No | No | No |

Criteria used to evaluate the benefits of genetic rescue included assessing whether the recipient population is isolated (based on hierarchical cluster analysis), whether the population is small ($N_a < 100$) and has been for multiple generations ($N_e < 100$), and whether F > 10%. Abbreviations: $N_e$, effective population size; *He*, expected heterozygosity; $N_a$, adult population size.

*Donor populations using the nearest neighbor are as follows (recipient/donor): Pacifica/Skyline; Skyline/Crystal Springs; Crystal Springs/Skyline; San Bruno/Skyline; Mindego/Pescadero; Pescadero/ Año Nuevo; Año Nuevo/Pescadero.

and adult abundance ($N_a$) (as an approximation of census size) among the sites sampled to assess whether populations in different parts of the range vary across these parameters. One of our major findings is that estimates of both $N_e$ and $N_a$ are small ($\leq 100$) for a majority of the sampled sites. Low empirical estimates for both $N_e$ and $N_a$ were more pronounced among the northern regional populations, with the exception of San Bruno (which we discuss below), where anthropogenic habitat alteration and landscape fragmentation have increasingly reduced connectivity among populations. Long-term maintenance of at least 10 populations with a minimum of 200 adults with equal sex ratios is a U.S. Fish and Wildlife Service recovery criterion for *T. s. tetrataenia* [26]. Of the seven focal sites we monitored, five have adult population size estimates below this recovery threshold and the smallest sites appear to have shifted sex ratios, suggesting that this goal has not been met. Furthermore, recent empirical studies suggest minimum $N_e$ thresholds for endangered species should be maintained above 100 to avoid short-term inbreeding depression and fitness loss and above 1000 to retain genetic diversity over longer time scales [64]. Our empirical findings ($N_e < 60$ in 6 of 7 focal sites) suggest that most populations are at risk of inbreeding depression and that more effective management of this species might be achieved by taking this into account. San Bruno was the only site that exceeded the minimum short-term threshold with an $N_e$ of 254, which was coupled with highest abundance estimate ($N_a = 1317$). Habitat enhancement activities over the past two decades at San Bruno may have facilitated the higher abundance estimates at this site despite being embedded in an urban matrix since the 1960s [35, 77], suggesting that habitat restoration and management may alleviate local loss of genetic diversity in this species.

Our results of the effective to census size ratio ($N_e/N$), where adult abundance ($N_a$) was used to approximate census size ($N$), are a tangible contribution towards conservation actions of this endangered snake. Given the connection of genetic and demographic processes in the effective to census size ratio ($N_e/N$) ratio, researchers have recommended using these parameters as interchangeable surrogates of each other [17, 80]. On the basis of theoretical expectations, effective population size ($N_e$) is typically smaller than census size ($N$), and $N_e/N$ ratios in real populations should range between 0.25–0.50 [18]. Comparing estimates for populations from a wide range of taxa, including some that are stable and some of conservation concern, Palstra & Ruzzante [80] reported empirical $N_e/N$ ratio estimates that ranged from 0.16 to 0.20, while slightly lower estimates (median values ~0.1) were recovered by Frankham [80]. Few studies have estimated $N_e/N$ ratios in snakes (especially across multiple populations) of a single species to document the magnitude of variation. However, Madsen et al. [82] reported effective size and census size estimates for an isolated population of adders (*Vipera berus*) that led to $N_e/N \approx 0.34$, and Bradke et al. [83] report similar $N_e/N$ ratios for two populations ($N_e/N = 0.27$ and 0.30) of the threatened Eastern massasauga (*Sistrurus catenatus*). Across our seven focal sites, $N_e/N$ ratios varied widely but generally ranged between 0.16–0.40; however, two estimates were much higher at Skyline and Pescadero (0.64 and 0.78 respectively). Several factors are known to influence the $N_e/N$ ratio at the population level, such as fluctuations in census size, unequal sex ratio and variance in reproductive success [81], so it is not surprising that we recovered a range of values across the different focal sites. We found some evidence that lower abundance might be associated with sex-ratios shifted towards more females, although this relationship was only significant if San Bruno (a site that has received intense habitat enhancements, as mentioned above) was treated as an outlier population ($R^2 = 0.55$, $p = 0.05$). Nonetheless, sex-ratios shifted towards females in smaller populations may be related to the divergent reproductive strategies of males and females of *T. sirtalis* [84–85]. Males typically invest more energy in mate-searching and courtship than females and this investment is likely to increase in populations with lower abundance, which may lead to reduced survival in males. However, other factors may also be at play. For some sites, the area of potential habitat was

larger than what could be adequately surveyed, which could lead to a lower census estimate relative to expectations based on the $N_e$ estimate from that same site (which does not rely on a thorough sampling of all individuals in the population). This might explain the higher ratios reported at Pescadero and Skyline. Nonetheless, these represent reasonable baseline estimates of the effective to census size ratio ($N_e/N$) in *T. s. tetrataenia* and can be used to evaluate temporal and spatial differences in $N_e/N$ ratios as future conservation actions are implemented.

Theoretical studies indicate that an ideal, genetically healthy population is one that has a sufficiently large $N_e$ coupled with moderate connectivity to other populations to maintain gene exchange and counter the effects of genetic drift [14]. When connectivity is disrupted, genetic differentiation between populations tends to increase and effective population sizes tend to decrease [16, 86]. We found greater population structure, lower $N_e$, and lower diversity within the northern regional populations than in the southern region, possibly because of the greater loss of physical and genetic connectivity due to anthropogenic disturbances over the last century (Fig 1). Genetic analyses of temporal datasets show that genetic differentiation has increased over time and that sites that are more isolated (Pacifica and San Bruno) show a greater change in differentiation over time. Pacifica in particular, had the lowest genetic diversity estimates of all sites sampled, small effective population size, the highest mean inbreeding coefficient (*F*), and decreased genetic diversity over time. Population surveys over the past several decades at this site have suggested that bottlenecks were likely the result of amphibian prey base declines from saltwater intrusion into wetlands from an eroded seawall during the 1980s [87–89]. Our genetic results also suggest that drift-mediated processes, as a function of small population size and reduced connectivity from neighboring populations, may also be contributing to decreasing trends in diversity at Pacifica.

## Concluding remarks

When contemporary disturbances are the driver of genetic differentiation, conservation strategies may focus on enhancing and reconnecting populations by reducing ecological threats and restoring habitat to pre-disturbance conditions [90]. Assisted gene flow, termed "genetic rescue," has also been a useful tool for slowing the decline of small, at-risk populations [91–94]. Following criteria developed by Frankham et al. [13], our evaluations of local population heterozygosity indicate that Pacifica may benefit from assisted gene flow from any other population in the northern region. San Bruno and Mindego also had marginally high inbreeding coefficients and both are isolated from other populations. However, for these sites the high abundance of adult snakes ($N_a \geq 200$) may buffer against the effects of genetic drift despite their isolation, given that the effects of drift are less pervasive in larger populations. Continued monitoring (both genetic and demographic) at these two sites may be necessary to assess population stability and whether assisted gene flow would help to retain genetic diversity over time.

Should a genetic management and recovery strategy be adopted for *T. s. tetrataenia*, the population genetic results provided here, along with estimates of abundance, can help to establish source and recipient populations. This information can be used with captive husbandry research programs to curb further loss of genetic diversity and strengthen fitness and (or) adaptive potential across the range of this remarkable gartersnake species. Although the risk of outbreeding depression is generally low between recently diverged populations [95], it is important to evaluate the potential loss of local adaptation for admixed populations [96]. Captive breeding efforts could be used to investigate whether outcrossing from the most divergent groups (northern versus southern regional groups) have favorable enough population-level fitness responses (e.g., increased genetic variation and fecundity) to outweigh concerns of

outbreeding depression [94, 97–98]. To assist assessment of status and trends in this subspecies, we also suggest continued demographic and genetic monitoring of the populations that were evaluated in this study. Genetic monitoring of any focal populations where genetic rescue efforts take place will also enable quantifying any genetic changes that may be associated with observable fitness effects (such as changes in population growth and size), assisted gene flow efforts, or further environmental change. Prior to our study there was little information published that pertained to genetic diversity and population demography across the range of *T. s. tetrataenia*. Our study shows how combining genetic and demographic monitoring of rare species can provide a more complete picture of the connectivity and viability of extant populations.

## Supporting information

**S1 Fig. Frequency histogram of distances moved by San Francisco garter snakes (*Thamnophis sirtalis tetrataenia*) between captures at five sites sampled in 2018.**
(PDF)

**S2 Fig. Structure assignments of individuals for K = 2–8 across all sites using the *T. s. tetrataenia* rangewide dataset.**
(PDF)

**S3 Fig. Structure assignment of individuals for *K* = 2 across all sites using the 12 putative outlier loci identified with BAYESCAN.** However, no meaningful patterns resulted from this analysis.
(PDF)

**S1 Table. Priors for covariate effects on capture probability for sites sampled, listed by years sampled.** For all sites we used a model that does not include individual heterogeneity in capture probability (p). Percentiles are lower and upper 95% Highest Posterior Density Interval limits.
(PDF)

**S2 Table. Pairwise genetic differentiation estimates using the 12 putative outlier loci identified with BAYESCAN for sampled sites with > 5 samples/site.** No values were significant using the $\alpha < 0.002$ after Bonferroni correction.
(PDF)

## Acknowledgments

We thank the California Department of Parks and Recreation, San Francisco Public Utilities Commission, Golden Gate National Recreation Area, San Francisco Recreation & Parks, Peninsula Open Space Trust, and Midpeninsula Regional Open Space District for providing access to their lands. We thank Natalie Reeder and Nixon Lam for access to San Bruno and for permission to share data. Many thanks to Elliot Schoenig for providing the photo of *T. s. tetrataenia* used in Fig 1 and to Tammy Lim for her time related to the 2004–2010 tissue organization. The manuscript benefitted much from comments by Jonathan Richmond, Shawna Zimmerman, and two anonymous reviewers. We thank Josh Hull (U.S. Fish and Wildlife Service) for his support of this research. We also wish to acknowledge the many agency staff, technicians, university students, and volunteers who assisted with capture-mark-recapture studies and collection of tissue samples, including Elizabeth Armistead, William Bauer, Rachael Burnham, Ryan Byrnes, Andrea Colton, Joie DeLeon, Julia Ersan, Jessica Gonzales, Ashley Estacio, Tianna Hanna, Richard Kim, John Kunna, Zach Leisz, Patrick Lien, William McCall, Daniel

Macias, Haley Mirtz, Nathan Moy, David Muth, Sean Parnell, Hailey Pexton, Natalie Reeder, Elliot Schoenig, Glenn Woodruff. Any use of trade, product, or firm names is for descriptive purposes only and does not imply endorsement by the U.S. government.

## Author Contributions

**Conceptualization:** Dustin A. Wood, Jonathan P. Rose, Brian J. Halstead, Amy G. Vandergast.

**Data curation:** Dustin A. Wood, Jonathan P. Rose, Ricka E. Stoelting.

**Formal analysis:** Dustin A. Wood, Jonathan P. Rose, Brian J. Halstead, Amy G. Vandergast.

**Funding acquisition:** Dustin A. Wood, Brian J. Halstead, Amy G. Vandergast.

**Investigation:** Dustin A. Wood, Jonathan P. Rose.

**Methodology:** Dustin A. Wood, Jonathan P. Rose, Brian J. Halstead.

**Project administration:** Brian J. Halstead.

**Resources:** Ricka E. Stoelting, Karen E. Swaim.

**Software:** Jonathan P. Rose.

**Validation:** Dustin A. Wood, Jonathan P. Rose.

**Visualization:** Dustin A. Wood.

**Writing – original draft:** Dustin A. Wood, Jonathan P. Rose, Brian J. Halstead, Amy G. Vandergast.

**Writing – review & editing:** Dustin A. Wood, Jonathan P. Rose, Brian J. Halstead, Ricka E. Stoelting, Karen E. Swaim, Amy G. Vandergast.

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
