## [Decision Letter · Decision Letter 0]

5 Feb 2020

PONE-D-19-34941

Combining genetic and demographic monitoring better informs conservation of an endangered urban snake

PLOS ONE

Dear Mr. Wood,

Thank you and your co-authors for submitting your manuscript to PLOS ONE. In this regard, three individuals were solicited to review this manuscript, and one declined. Two reviewers who provided reviews recognized the values, both narrow and broad, of this submission and each appreciated the statistical rigor. While one suggested "Accept," and the other "Minor Revision," both were in agreement that little needs to be done for this manuscript to be published in PLOS ONE.

Thus, when these reviews were considered by the Academic AE, their implementation is adjudicated as minor, and the formal decision is thus "Accept Following Minor Revisions." The two reviewers provide some minor points of consideration as well as some more esoteric considerations for the discussion (with specifics provided in the individual reviews). If the authors consider these recommendations, document (in a response letter) the changes deemed appropriate, rebut those deemed inappropriate, and resubmit the manuscript, the AE will recommend Acceptance without sending out for further review.

Thank you for submitting this manuscript to PLOS ONE. I look forward to reviewing a revised manuscript in the very near future.

We would appreciate receiving your revised manuscript by Mar 21 2020 11:59PM. To enhance the reproducibility of your results, we recommend that if applicable you deposit your laboratory protocols in protocols.io, where a protocol can be assigned its own identifier (DOI) such that it can be cited independently in the future. For instructions see: http://journals.plos.org/plosone/s/submission-guidelines#loc-laboratory-protocols

We look forward to receiving your revised manuscript.

Kind regards,

Mark A. Davis, Ph.D.

Academic Editor

PLOS ONE

Additional Editor Comments (if provided):

This manuscript provides an interesting, unique, and rigorous exploration of a western North American snake in an urban context. The paper is well written, well executed, and the authors to be commended. Three reviews were solicited, and two agreed to provide comments. Both reviewers were in agreement that the manuscript is both technically sound and germane to the volume. And while I have not provided a substantive review, I have read the manuscript and agree with their recommendations.

Journal Requirements:

2. In your Methods section, please provide additional location information of the study points, including geographic coordinates for the data set if available.

3. Our internal editors have looked over your manuscript and determined that it is within the scope of our Biodiversity Conservation Call for Papers. This collection of papers is headed by a team of Guest Editors for PLOS ONE (https://collections.plos.org/s/biodiversity). The Collection will encompass a diverse range of research articles on biodiversity conservation, including the impacts of habitat fragmentation and drivers of extinction risk. Additional information can be found on our announcement page: https://collections.plos.org/s/biodiversity

If you would like your manuscript to be considered for this collection, please let us know in your cover letter and we will ensure that your paper is treated as if you were responding to this call. If you would prefer to remove your manuscript from collection consideration, please specify this in the cover letter.

4. Thank you for stating the following in the Competing Interests/Financial Disclosure * (delete as necessary) section:

"Funding for this project was provided by U.S. Geological Survey and U.S. Fish and Wildlife Service, and Swaim Biological, Inc (as in-kind services)"

We note that you received funding from a commercial source: Swaim Biological, Inc.

6. Your ethics statement must appear in the Methods section of your manuscript. If your ethics statement is written in any section besides the Methods, please move it to the Methods section and delete it from any other section. Please also ensure that your ethics statement is included in your manuscript, as the ethics section of your online submission will not be published alongside your manuscript.

7. We note that Figures 1 and 6 in your submission contain map images which may be copyrighted. All PLOS content is published under the Creative Commons Attribution License (CC BY 4.0), which means that the manuscript, images, and Supporting Information files will be freely available online, and any third party is permitted to access, download, copy, distribute, and use these materials in any way, even commercially, with proper attribution. For these reasons, we cannot publish previously copyrighted maps or satellite images created using proprietary data, such as Google software (Google Maps, Street View, and Earth). For more information, see our copyright guidelines: http://journals.plos.org/plosone/s/licenses-and-copyright.

You may seek permission from the original copyright holder of Figures 1 and 6 to publish the content specifically under the CC BY 4.0 license. 

If you are unable to obtain permission from the original copyright holder to publish these figures under the CC BY 4.0 license or if the copyright holder’s requirements are incompatible with the CC BY 4.0 license, please either i) remove the figure or ii) supply a replacement figure that complies with the CC BY 4.0 license. Please check copyright information on all replacement figures and update the figure caption with source information. If applicable, please specify in the figure caption text when a figure is similar but not identical to the original image and is therefore for illustrative purposes only.

Reviewers' comments:

Reviewer's Responses to Questions

**Comments to the Author**

1. Is the manuscript technically sound, and do the data support the conclusions?

Reviewer #1: Yes

Reviewer #2: Yes

2. Has the statistical analysis been performed appropriately and rigorously? 

Reviewer #1: Yes

Reviewer #2: Yes

3. Have the authors made all data underlying the findings in their manuscript fully available?

Reviewer #1: Yes

Reviewer #2: Yes

4. Is the manuscript presented in an intelligible fashion and written in standard English?

Reviewer #1: Yes

Reviewer #2: Yes

5. Review Comments to the Author

Reviewer #1: This paper represents a rigorous application of advanced genetic and demographic analyses in the context of a protected species and will contribute greatly to conservation of the focal subspecies. Data are presented in a clear and concise manner and all methods and results are explained fully. While not a consideration for publication in PLoS ONE, the topic and results of the study are both broadly and specifically interesting, presenting a valuable example of the use of the methods herein that can be applied elsewhere while also having practical implications for the management of a specific species.

I am recommending that the paper be accepted without revision. If revised, I provide the following recommendations and corrections.

Materials and Methods

Field methods and sample selection

Line 128 - describe tissue sampling. What samples were collected and how?

Line 128 - explain focal sites. How/why were focal sites chosen? How do focal sites differ from "satellite" sites? Is it just sample size?

Throughout - how do the sites compare to one another? In size? In habitat? In sampling coverage? It is mentioned in the discussion that some sites are much larger than the area that was sampled. Additional information to specify for which sites this was true and how it might affect the results or interpretation of the results would be helpful.

A few grammatical errors:

Line 401 ..."filters we applied..." should be filters were applied

Line 411 ..."where it was assigned as distinct genetic cluster..." should be assigned as a distinct genetic cluster

Line 426 ..."each regional clusters that supported..." clusters should be cluster

Reviewer #2: Wood et al. survey numerous sites for Thamnophis sirtalis tetrataenia and generate a RAD-seq data set for this taxon to better understand how best to implement future conservation efforts. I think that this is an interesting and important data set that has been thoroughly analyzed and has real world conservation applications. I have only a few comments listed below.

The sampling dates to look at temporal change in population genetic structure are 2004-2010 and 2015- 2018. How were these thresholds for change over time decided on? It seems arbitrary and not consistent across all sites.

For the hierarchical structure analyses, was delta-K used in subsequent structure runs? If so, how was K=1 assessed (it is a known issue that with this particular method K=1 cannot be assessed)? Also I am assuming that the hierarchical structure analyses were run until there was no additional population structure? A bit more details would be good here.

L451 – I’m not entirely sold that this method is highlighting urban development as influencing population structure. Perhaps more details on how this is the case would clarify this. Otherwise, I think it’s a bit overstated.

That’s super interesting that T. s. tetrataenia is paraphyletic given how morphologically different it is!

What’s the longevity of these snakes – are the authors certain that in some of these small populations that the same individuals weren’t sampled in both time periods used for temporal changes in genetic structure? E.g., in Pescadero there’s only 6 years between sampling periods, presumably these garters are living 6+ years and the small population sizes at these sites might make it more likely to sample the same individuals after six years.

Were these tissue samples accessioned at a natural history collection? I really like the temporal sampling, and in the discussion the authors state that future studies could build on this study but if a different approach to generating sequence data was used it wouldn’t be possible. Perhaps I missed it, but I would really like to see these samples available.

6. PLOS authors have the option to publish the peer review history of their article (what does this mean?). If published, this will include your full peer review and any attached files.

Reviewer #1: No

Reviewer #2: No

---

## [Author Response · Author response to Decision Letter 0]

30 Mar 2020

The below responses to the reviewers was also attached with the manuscript.

Response to PlosOne Journal Reviews

Reviewer #1: This paper represents a rigorous application of advanced genetic and demographic analyses in the context of a protected species and will contribute greatly to conservation of the focal subspecies. Data are presented in a clear and concise manner and all methods and results are explained fully. While not a consideration for publication in PLoS ONE, the topic and results of the study are both broadly and specifically interesting, presenting a valuable example of the use of the methods herein that can be applied elsewhere while also having practical implications for the management of a specific species.

I am recommending that the paper be accepted without revision. If revised, I provide the following recommendations and corrections.

Materials and Methods

Field methods and sample selection

Line 128 - describe tissue sampling. What samples were collected and how?

Response to Reviewer 1: We added information regarding what type of tissue was taken and how this was conducted on page 6, lines 154-156.

Line 128 - explain focal sites. How/why were focal sites chosen? How do focal sites differ from "satellite" sites? Is it just sample size?

Response to Reviewer 1: We added text to help clarify the difference between focal & satellite sites with the section “Field methods and sample collection”. This can be found on page 6-7, lines 150-196. The focal sites were where we focused the demographic studies whereas satellite sites were not involved in demographic analyses and had much smaller sample sizes. 

Throughout - how do the sites compare to one another? In size? In habitat? In sampling coverage? It is mentioned in the discussion that some sites are much larger than the area that was sampled. Additional information to specify for which sites this was true and how it might affect the results or interpretation of the results would be helpful.

Response to Reviewer 1: We have added estimates of the area effectively sampled by our traps for each site. Based on data we have on the distances moved by San Francisco gartersnakes between captures (97% of snake movements were < 200 m), we calculated the effective sample area for each site by buffering traps by 200 m and summing the area of habitat sampled by traps at each site. We have also added estimates of the total area of habitat available at each site (adding information in Table 1 and a supplemental figure, S1 Fig), so readers can compare how much of the habitat at each site was effectively sampled for San Francisco gartersnakes. The following text has been added to the methods (pages 8-9, lines 244–277):

“Sites differed in size of available habitat and in the area sampled by traps and cover objects. To define the total area of available habitat for each site, we created polygons in ArcGIS version 10.7.1 [33] that encompassed all suitable habitat, whether wetlands or non-forested uplands. The effective area sampled was then calculated by using a fixed buffer of 200 m around all trap and artificial cover object locations for each site. We chose a 200 m buffer based on the maximum distance moved between captures for greater than 95 percent of individual T. s. tetrataenia at our study sites (S1 Fig). The total site area and effective area sampled for each site are given in Table 1. At Pacifica and San Bruno, nearly all of the available habitat for T. s. tetrataenia was within the area effectively sampled by traps, and the nearest known population of T. s. tetrataenia was more than 2.5 km away. In contrast, although the area sampled at Skyline and Crystal Springs was comparable to Pacifica, only 60-70 percent of the area was sampled because suitable wetland habitat was present nearby and additional habitat (not included in our calculations) is present along most of the 7 km corridor separating these two sites. Año Nuevo, Mindego, and Pescadero were much larger sites and the area sampled was less than 60 percent of the available habitat.”

A few grammatical errors:

Line 401 ..."filters we applied..." should be filters were applied

Response to Reviewer 1: We fixed the grammatical error.

Line 411 ..."where it was assigned as distinct genetic cluster..." should be assigned as a distinct genetic cluster

Response to Reviewer 1: We fixed the grammatical error.

Line 426 ..."each regional clusters that supported..." clusters should be cluster

Response to Reviewer 1: We fixed the grammatical error.

Reviewer #2: Wood et al. survey numerous sites for Thamnophis sirtalis tetrataenia and generate a RAD-seq data set for this taxon to better understand how best to implement future conservation efforts. I think that this is an interesting and important data set that has been thoroughly analyzed and has real world conservation applications. I have only a few comments listed below.

The sampling dates to look at temporal change in population genetic structure are 2004-2010 and 2015- 2018. How were these thresholds for change over time decided on? It seems arbitrary and not consistent across all sites.

Response to Reviewer 2: This was an opportunistic use of previously collected, older samples that were available. We decided there were two breaks in time between concerted tissue sampling (1) an earlier temporal sample with a wider range of dates from 2004-2010 depending on the site and (2) a later temporal sample across a three year span 2015-2018. 

For the hierarchical structure analyses, was delta-K used in subsequent structure runs? If so, how was K=1 assessed (it is a known issue that with this particular method K=1 cannot be assessed)? Also I am assuming that the hierarchical structure analyses were run until there was no additional population structure? A bit more details would be good here.

Response to Reviewer 2: Yes, we used delta-K in subsequent runs for STRUCTURE analyses. As we mentioned in the methods, we evaluated the clustering results of our STRUCTURE analyses using an independent analysis of population structure, discriminant analysis of principal components (DAPC). This analysis differs from STRUCTURE analysis in that it does not require the assumptions of HWE. Given that we recovered the same clusters that were observed in the STRUCTURE analysis, cluster results above K=1 are much more likely. In addition, the results from the mean log likelihood [lnP(D|K] score for each K value up to 7 or 8 had higher likelihood values than K at 1 (See Supporting information S2 Figure). 

L451 – Iʼm not entirely sold that this method is highlighting urban development as influencing population structure. Perhaps more details on how this is the case would clarify this. Otherwise, I think itʼs a bit overstated.

Response to Reviewer 2: We removed the phrase “due to urban development” from the sentence. The sentence now reads as “In northern San Mateo County, cells separating San Bruno from other sites received “hotter’ values of genetic discontinuity, highlighting the isolation of individuals at this site.” See page 22, lined 639-641.

Thatʼs super interesting that T. s. tetrataenia is paraphyletic given how morphologically different it is!

Whatʼs the longevity of these snakes – are the authors certain that in some of these small populations that the same individuals werenʼt sampled in both time periods used for temporal changes in genetic structure? E.g., in Pescadero thereʼs only 6 years between sampling periods, presumably these garters are living 6+ years and the small population sizes at these sites might make it more likely to sample the same individuals after six years.

Response to Reviewer 2: For most sites, the time between tissue sample collection was greater than 10 years, and as Reviewer 2 noted, only Pescadero was shorter at 6 years. We do not know the actual generation time of Thamnophis sirtalis but an average generation time for gartersnakes is approximately 3-4 years. Only a limited number of studies on survival in T. s. tetrataenia have been conducted, but they have shown that under ideal conditions only 50% of neonates survive after 2 years of age and only 2% of neonates survive to age 5 (Barry 1996). So our temporal samples are likely sampling different generations with a low likelihood of individuals present in both temporal samples. In addition, tissue samples were taken using tail tips and if a snake was captured with a blunt tail, then this was a good indication that a tissue sample had already been collected. Additionally, as mentioned in the Methods section we marked individuals at all sites with a unique ventral brand and used Passive Integrated Transponders (PIT) tags. When an individual was recaptured, we would not take a second tissue sample. Lastly, we did look for duplicate genotypes in our dataset and did not find any. 

Were these tissue samples accessioned at a natural history collection? I really like the temporal sampling, and in the discussion the authors state that future studies could build on this study but if a different approach to generating sequence data was used it wouldnʼt be possible. Perhaps I missed it, but I would really like to see these samples available.

Response to Reviewer 2: The San Diego Genetic Facility at USGS houses genetic materials that are part of active scientific investigations. The Smithsonian is the official repository for USGS samples upon completion of active research. The Sundry Civil Act of March 3, 1879 (20 U.S.C. 59), as amended, directs that “All collections of rocks, minerals, soils, fossils, and objects of natural history, archaeology, and ethnology, made by the National Ocean Survey, the [United States] Geological Survey, or by any other parties for the Government of the United States, when no longer needed for investigations in progress shall be deposited in the National Museum [Smithsonian Institution National Museum of Natural History].

---

## [Editor Report · Decision Letter 1]

7 Apr 2020

Combining genetic and demographic monitoring better informs conservation of an endangered urban snake

PONE-D-19-34941R1

Dear Dr. Wood,

We are pleased to inform you that your manuscript has been judged scientifically suitable for publication and will be formally accepted for publication once it complies with all outstanding technical requirements.

With kind regards,

Mark A. Davis, Ph.D.

Academic Editor

PLOS ONE

Additional Editor Comments (optional):

The original manuscript was well done, and this revision manages to improve upon the original's excellence. It is incredibly well-written, and of the highest statistical rigor. It will make a fantastic contribution to the Biodiversity Conservation Collection, and I commend the authors for their work
---

## [Editor Report · Acceptance letter]

9 Apr 2020

PONE-D-19-34941R1 

Combining genetic and demographic monitoring better informs conservation of an endangered urban snake 

Dear Dr. Wood:

I am pleased to inform you that your manuscript has been deemed suitable for publication in PLOS ONE. Congratulations! Your manuscript is now with our production department. 

With kind regards,

on behalf of

Dr. Mark A. Davis 

Academic Editor

PLOS ONE